# Beneficial Effects of Essential Oils from the Mediterranean Diet on Gut Microbiota and Their Metabolites in Ischemic Heart Disease and Type-2 Diabetes Mellitus

**DOI:** 10.3390/nu14214650

**Published:** 2022-11-03

**Authors:** María José Sánchez-Quintero, Josué Delgado, Dina Medina-Vera, Víctor M. Becerra-Muñoz, María Isabel Queipo-Ortuño, Mario Estévez, Isaac Plaza-Andrades, Jorge Rodríguez-Capitán, Pedro L. Sánchez, Maria G. Crespo-Leiro, Manuel F. Jiménez-Navarro, Francisco Javier Pavón-Morón

**Affiliations:** 1Instituto de Investigación Biomédica de Málaga y Plataforma en Nanomedicina (IBIMA-Plataforma BIONAND), 29590 Málaga, Spain; 2Unidad de Gestión Clínica Área del Corazón, Hospital Universitario Virgen de la Victoria, 29010 Málaga, Spain; 3Centro de Investigación Biomédica en Red de Enfermedades Cardiovasculares (CIBERCV), Instituto de Salud Carlos III, 28029 Madrid, Spain; 4Higiene y Seguridad Alimentaria, Facultad de Veterinaria, IPROCAR, Universidad de Extremadura, 10003 Cáceres, Spain; 5Unidad de Gestión Clínica de Salud Mental, Hospital Regional Universitario de Málaga, 29010 Málaga, Spain; 6Departamento de Dermatología y Medicina, Facultad de Medicina, Universidad de Málaga (UMA), 29010 Málaga, Spain; 7Unidad de Gestión Clínica Intercentros de Oncología Médica, Hospitales Universitarios Regional y Virgen de la Victoria y Centro de Investigaciones Médico Sanitarias (CIMES), 29010 Málaga, Spain; 8Departamento de Especialidades Quirúrgicas, Bioquímica e Inmunología, Facultad de Medicina, Universidad de Málaga (UMA), 29010 Málaga, Spain; 9Instituto Universitario de Investigación de Carne y Productos Cárnicos (IPROCAR), Universidad de Extremadura (UEX), 10003 Cáceres, Spain; 10Servicio de Cardiología, Hospital Universitario de Salamanca, Universidad de Salamanca, Instituto de Investigación Biomédica de Salamanca (IBSAL), 37007 Salamanca, Spain; 11Servicio de Cardiología, Complexo Hospitalario Universitario A Coruña (CHUAC), Universidade da Coruña (UDC), Instituto Investigación Biomédica A Coruña (INIBIC), 15006 A Coruña, Spain

**Keywords:** carbonyl, cytokine, chemokine, nutraceutical, parsley, pentosidine, prebiotic, protein carbonyl, rosemary, savory, short-chain fatty acid, trimethylamine *N*-oxide

## Abstract

Ischemic heart disease (IHD) and type-2 diabetes mellitus (T2DM) remain major health problems worldwide and commonly coexist in individuals. Gut microbial metabolites, such as trimethylamine *N*-oxide (TMAO) and short-chain fatty acids (SCFAs), have been linked to cardiovascular and metabolic diseases. Previous studies have reported dysbiosis in the gut microbiota of these patients and the prebiotic effects of some components of the Mediterranean diet. Essential oil emulsions of savory (*Satureja hortensis*), parsley (*Petroselinum crispum*) and rosemary (*Rosmarinus officinalis*) were assessed as nutraceuticals and prebiotics in IHD and T2DM. Humanized mice harboring gut microbiota derived from that of patients with IHD and T2DM were supplemented with L-carnitine and orally treated with essential oil emulsions for 40 days. We assessed the effects on gut microbiota composition and abundance, microbial metabolites and plasma markers of cardiovascular disease, inflammation and oxidative stress. Our results showed that essential oil emulsions in mice supplemented with L-carnitine have prebiotic effects on beneficial commensal bacteria, mainly *Lactobacillus* genus. There was a decrease in plasma TMAO and an increase in fecal SCFAs levels in mice treated with parsley and rosemary essential oils. Thrombomodulin levels were increased in mice treated with savory and parsley essential oils. While mice treated with parsley and rosemary essential oils showed a decrease in plasma cytokines (INFɣ, TNFα, IL-12p70 and IL-22); savory essential oil was associated with increased levels of chemokines (CXCL1, CCL2 and CCL11). Finally, there was a decrease in protein carbonyls and pentosidine according to the essential oil emulsion. These results suggest that changes in the gut microbiota induced by essential oils of parsley, savory and rosemary as prebiotics could differentially regulate cardiovascular and metabolic factors, which highlights the potential of these nutraceuticals for reducing IHD risk in patients affected by T2DM.

## 1. Introduction

Ischemic heart disease (IHD) is the most common form of cardiovascular disease and the leading cause of death in Europe in recent years in both men and women [1,2], excluding the emergence of COVID-19. In past decades, various strategies have been established to reduce the morbidity and mortality of patients with coronary disease, and important improvements have been achieved. Type-2 diabetes mellitus (T2DM) often leads to an increased risk of development of several forms of cardiovascular complications, such as IHD [3]. Previous studies in patients with IHD suggest that the presence of T2DM is linked to an impairment of the immune system mediated by the gut microbiota, finding significantly less beneficial or commensal bacteria in these patients [4]. In fact, it has been reported that T2DM can significantly alter intestinal microbial populations in patients with IHD [4]. Recently, there is a growing interest in replacing pharmaceutical therapies by dietary interventions based on natural bioactive food components, which has led to the development of new alternative therapies to extend and improve the quality of life of patients. On this line, plant-derived bioactive components, such as essential oils, have been studied as potential modulators of physiological processes related to IHD.

Growing evidence has demonstrated a link between gut dysbiosis and cardiovascular disease risk [5,6], and that the modulation of the gut microbiome may contribute to ameliorating these diseases. Inhibitory substances of non-beneficial microbiota can be present in spices or plants commonly used in the Mediterranean diet [7]. However, the consumption of usual levels from these plants within the diet would have a limited beneficial effect on bacteria population. Therefore, it would be necessary to use them as nutraceuticals, understanding this term as a food that provides health benefits, including the prevention and/or treatment of diseases. Furthermore, different studies have shown that essential oils affect both the microbial population composition and its activity, which have a direct effect on IHD [8]. Thus, these substances may be used to regulate the immune response and enhance the presence of beneficial microorganisms, such as *Lactobacillus* spp., to the detriment of those bacteria that do not provide benefits or produce unwanted metabolites [9].

One example of how gut microbiota has an impact on IHD is through the formation of trimethylamine *N*-oxide (TMAO). TMAO is a pro-atherogenic substance which has been associated with the development of adverse cardiovascular events in the general population over the last several years [10,11,12,13]. TMAO results from the oxidation of trimethylamine (TMA) in the liver, which is in turn produced by certain colonic microbial populations, such as some species of Bacteroidetes and Firmicutes [4]. These common microorganisms contribute to TMA formation in the colon through the precursors L-carnitine and phosphatidylcholine [14]. Previous studies have observed that plant origin substances can act as prebiotics and reduce the transformation of L-carnitine to TMA through their action on the colonic microbiota [15]. Considering this, the gut microbiota could be modulated to restore the balance between beneficial and detrimental microorganisms regarding TMA and TMAO production.

Another mechanism by which gut microbiota is related to IHD is through the formation of short-chain fatty acids (SCFAs). Dietary fibers, resistant starch and other polysaccharides that evade digestion by host enzymes in the upper gut are metabolized by the microbiota in the cecum and colon, through anaerobic fermentation [16,17]. Their major end-products are SCFAs, monocarboxylic acids with two to five carbons [18,19]. Acetic, propionic and butyric acids are the main SCFAs at the rate of 60:20:20, respectively, and are generated in the human gut daily, depending on the fiber content of the diet and microbiota structure [20,21,22,23,24]. The rate between these fatty acids has consequences in terms of gut health, adiposity and IHD. Butyric acid is used as an energy source by enterocytes and strengthens the intestinal epithelium; while acetic acid positively affects body weight control, insulin sensitivity and control of appetite; and propionic acid decreases liver lipogenesis, hepatic and serum cholesterol levels, preventing or hampering the IHD [23,25,26].

Besides the effects of essential oils on microbiota, these compounds may be related to IHD through many other mechanisms. For instance, the capacity of certain plant origin products to prevent the formation of free radicals that would interact with cellular DNA, confer antioxidant, antifungal, immunomodulatory, anticancer and anti-inflammatory activity in animal models [27,28]. In fact, higher inflammation levels are usually reported in patients suffering from cardiovascular disease and/or T2DM than those found in healthy ones [29]. Thus, extensive studies aiming to counteract these diseases have shown the potent therapeutic effects of plant-derived compounds against a series of chronic diseases, such as cardiovascular disease and T2DM, unraveling new anti-inflammatory and antioxidant molecules [30]. The accretion of oxidized proteins in tissues is a pathological hallmark of chronic diseases, such as those aforementioned, and hence, the protection of plant antioxidants against protein oxidation may be a plausible means to inhibit the biological consequences of protein oxidation: cell dysfunction and health disorders [31]. The formation of protein carbonyl and pentosidine is increased under a variety of stress conditions, especially oxidative stress and inflammation, contributing to severe vascular and cardiovascular complications [32]. Carbonyls are the result of a mechanism by which sensible alkaline amino acids (lysine, proline, arginine) undergo oxidative deamination to yield aldehydes (α-aminoadipic semialdehyde (AAS), also known as allysine, and ɣ-glutamic semialdehyde (GGS)) [33]. The formation of AAS in the presence of glucose and toxic diabetes metabolites suggests that it may be formed under pathological pro-diabetic conditions [34]. In fact, AAS and its final oxidation product, the α-aminoadipic acid, are commonly used as markers of many chronic diseases involving enduring oxidative stress, such as T2DM [31,35].

In this study, we generated a gnotobiotic murine model carrying colonic microbiota derived from patients with IHD and T2DM. The objective of this work was to evaluate the effect of the administration of essential oils from commonly used herbs in the Mediterranean diet on the gut microbial population, TMAO and SCFA levels, inflammation and oxidative stress, using this gnotobiotic murine model.

## 2. Materials and Methods

### 2.1. Ethical Statement and Animals

This study was designed in accordance with the European directive 2010/63/EU for the protection of animals used for scientific purposes and the Spanish regulations for the care and use of laboratory animals (RD53/2013 and RD118/2021). All protocols and experiments were approved by the “*Animal Experimentation Ethics Committee of BIONAND*”, Málaga, Spain (23/10/2018/151). Randomly cycling female CD1 mice (Janvier Labs, Le Genest-Saint-Isle, France) were acclimated for 4 weeks and housed in a humidity and temperature-controlled vivarium and maintained on a 12-h light-dark cycle with food (Appendix A) and water *ad libitum*.

### 2.2. Humanized Gnotobiotic Mouse Model

A humanized gnotobiotic mouse model was developed through the fecal microbiota transplantation of human feces into mice. Prior to the transplantation procedure, 18-week-old mice (*n* = 40) were treated with broad-spectrum antibiotics for 10 consecutive days to induce gut microbiota depletion, combining both oral gavage administration and drinking water supplementation. Thus, mice were orally treated by gavage every 12 h with an antibiotic cocktail (10 mL kg^−1^ body weight) consisting of vancomycin (5 mg mL^−1^), neomycin (10 mg mL^−1^) and metronidazole (10 mg mL^−1^), and ampicillin was administered in drinking water (1 g L^−1^) [36,37,38]. Then, mice received a fecal microbiota transplantation derived from a selection of patients to recolonize the gastrointestinal tract, as previously described [38]. Specifically, three patients with IHD, T2DM and high plasma levels of TMAO were eligible for feces donation. Fresh fecal samples were suspended in sterile phosphate buffered saline (PBS), and glycerol was added at 10% (*v*/*v*) final concentration; then, samples were aliquoted into cryotubes and immediately stored at −80 °C. At the moment of the transplantation, aliquot samples from these patients were thawed in ice and mixed in equal quantities as a sample pool to guarantee that every mouse received a similar microbiota load. The sample pool was centrifuged for 2 min at 2800× *g*, and the supernatant was transferred to mice by oral gavage for three consecutive days, according to previously published protocols [38,39,40]. Mice were moved to clean cages every two days to minimize the coprophagia and to prevent reinoculation from old feces.

### 2.3. Study Design and Treatments

#### 2.3.1. Experimental Groups

Transplanted mice were randomly assigned to five experimental groups, with eight animals per group (*n* = 8), in a cage at the beginning of the study, based on a previous similar study [15]: (1) control group, mice without L-carnitine supplementation or essential oil administration; (2) carnitine group, mice with L-carnitine supplementation but not essential oil administration; (3) savory group, mice with L-carnitine supplementation and savory essential oil; (4) parsley group, mice with L-carnitine supplementation and parsley essential oil; and (5) rosemary group, mice with L-carnitine supplementation and rosemary essential oil. No criteria were used for including and excluding animals during the experiment.

#### 2.3.2. Preparation of Essential Oil Emulsions

Three commercial essential oils derived from commonly used herbs in the Mediterranean diet were purchased from Farmacia Rico Néstares (Málaga, Spain): savory (*Satureja hortensis*), parsley (*Petroselinum crispum*) and rosemary (*Rosmarinus officinalis*). For oral administration, oil-in-water (O/W) emulsions were prepared with these essential oils at 3.38% (*v*/*v*) concentration in distilled water with lecithin (0.76%, *w*/*v*) and maltodextrin (21.47%, *w*/*v*) as emulsifiers to prevent the oil and water phases separation.

#### 2.3.3. Treatment with Essential Oil Emulsions

During a total of 40 days, 20-week-old female mice from four groups (i.e., carnitine, savory, parsley and rosemary groups) were given drinking water supplemented with L-carnitine (0.02%, *w*/*v*), a precursor of TMA to stimulate the TMAO production, and treated with essential oil emulsions or vehicle emulsion (i.e., treatment with a vehicle solution containing distilled water, lecithin and maltodextrin in the same proportions but without essential oils). In contrast, mice from the control group were given drinking water without L-carnitine. The essential oils were administered daily by gavage at 100 mg kg^−1^ for 40 days after considering their spectrum of action and effective dose [41]. Following the same procedure, mice from the control and carnitine groups were administered daily by gavage with the vehicle emulsion. To avoid any confounder, treatments were always applied in the same order of groups, relocating the mice in a clean cage until the last mouse was treated. However, four mice died during the treatment period and the number of animals per group at the end of treatments was as follows: (1) control group, *n* = 7; (2) carnitine group, *n* = 7; (3) savory group, *n* = 8; (4) parsley group, *n* = 8; and (5) rosemary group, *n* = 6.

### 2.4. Collection of Fecal and Plasma Samples

Twenty-four hours after the last administration of essential oil emulsions or vehicle emulsion, mice were euthanized by decapitation, and trunk blood samples (0.8–1.0 mL) were collected in Microvette CB300 plasma/lithium heparin (Thermo Fisher Scientific, Waltham, MA, USA) tubes with volumes up to 300 μL. Blood samples were centrifuged at 3500× *g* for 5 min and the supernatant plasmas were stored at −80 °C until determinations were performed. In addition, fresh fecal samples were collected directly into the small tubes and stored at −80 °C until.

### 2.5. Microbiota Analysis by 16S rRNA Gene Sequencing

Fecal microbiota was analyzed after essential oil treatments. Briefly, total DNA was isolated from feces using the QIAamp DNA Stool Mini Kit (Qiagen, Hilden, Germany). The Ion 16S Metagenomics Kit (Thermo Fisher Scientific, Waltham, MA, USA) was used to amplify the 16S rRNA gene region from stool DNA using two primer pools (V2-4-8 and V3-6, 7-9) covering hypervariable regions of the 16S rRNA region in bacteria. The Ion Plus Fragment Library Kit (Thermo Fisher Scientific, Waltham, MA, USA) was used to ligate barcoded adapters to the generated amplicons and create the barcoded libraries. Template preparation of the created amplicon libraries was performed on the automated Ion Chef System using the Ion 520^TM^/530^TM^ Kit-Chef (Thermo Fisher Scientific, Waltham, MA, USA), according to the manufacturer’s instructions. Sequencing was carried out on an Ion 520 chip using the Ion S5^TM^ System (Thermo Fisher Scientific, Waltham, MA, USA).

### 2.6. L-carnitine, TMA and TMAO Levels in Plasma

A total volume of 15 µL of plasma was incubated with 45 µL of cold methanol for 2 h at −80 °C. After incubation, samples were centrifuged at 18,200× *g* for 12 min at 4 °C, and supernatant was collected and stored at −80 °C for further analysis. L-carnitine, TMA and TMAO levels were quantified using a Dionex UltiMate 3000 RSLC system coupled to Q Exactive Hybrid Quadrupole-Orbitrap Mass Spectrometer (Thermo Fisher Scientific, Waltham, MA, USA). An Accucore HILIC 150 × 2.1 mm × 2.6 µm column was used as a stationary phase, while as a mobile phase we used a solvent (A: H_2_O with 0.005 M ammonium formate pH 4.88; and B: acetonitrile and H_2_O (9:1) with 0.005 M ammonium formate pH 4.9). The gradient was isocratic 70% (A/B), the injection volume was 5 µL and flow rate was set as 0.4 mL min^−1^. Detection of L-carnitine, TMA and TMAO was performed with positive ionization in full scan with 70,000 full width at half maximum (FWHM), using the ions *m*/*z* 162.1125, 60.0808 and 76.0757 as targets, respectively, and setting 5 ppm of accuracy. Retention time was 2.08 ± 0.1 for L-carnitine, 2.3 ± 0.1 min for TMA and 2.03 ± 0.1 min for TMAO. Serial dilutions of commercial standards for both metabolites from Sigma-Aldrich (Merck KGaA, Darmstadt, Germany) were used to determine the linearity. Limit of detection (LOD) and limit of quantitation (LOQ) for TMA and TMAO were calculated at the lowest evaluable concentration level at which the qualifier ion signal exceeds the noise level by factor of 10 for LOD and 3.5 for LOQ: (1) L-carnitine, TMA: LOD = 0.2 ng mL^−1^ and LOQ = 0.6 ng mL^−1^, LOD = 0.225 ng mL^−1^ and LOQ = 0.7 ng mL^−1^; and (2) TMAO, LOD = 0.175 ng mL^−1^ and LOQ = 0.5 ng mL^−1^.

### 2.7. SCFA Species in Feces

Acetic, propionic and butyric acids were determined in the feces of mice treated with essential oil emulsions or vehicle emulsion. Feces were weighted and treated with 3 mL of double deionized water and hexane 50% (*v*/*v*), vortexed and sonicated for 5 min in a bath. Finally, samples were centrifuged at 1400× *g* for 5 min. To measure acetic, butyric and propionic acid, a small volume from the upper phase was injected in a 6890 N Flame Ionization Detector Gas Chromatograph System equipped with a DB-WAX 60 × 0.32 mm × 0.25 µm column (Agilent, Santa Clara, CA, USA). The temperature of injector and detector was set starting at 100 °C and increasing up to 250 °C in 15 min. The retention time of these metabolites were 3.58 ± 0.05, 4.18 ± 0.05 and 4.93 ± 0.05 min for acetic, propionic and butyric acids, respectively. The standards used were purchased from Dr. Ehrenstorfer (LGC Standards, Middlesex, UK). SCFA species were expressed as mg g^−1^ feces.

### 2.8. Determination of Other Analytes in Plasma

#### 2.8.1. Cardiovascular Markers-

Markers for cardiovascular disease were measured using the Mouse CVD Magnetic Bead Panel 1 (Merck Millipore, Burlington, MA, USA), according to manufacturer’s instructions. A panel of seven cardiovascular markers was measured for this study using a 96-well plate: sE-Selectin, sICAM-1, Pecam-1, sP-Selectin, PAI-1 (total), proMMP-9 and Thrombomodulin. All samples were prepared following the protocol and were run in duplicate in a 96-well plate. Briefly, a total volume of 25 mL of 1:20 diluted sample was mixed with 25 μL of Mixed Beads. The plate was incubated with agitation on a plate shaker overnight. After incubation, 25 μL of detection antibodies were added to wells, and the plate was incubated with agitation for 1 h at room temperature before adding 25 μL of Streptavidin-Phycoerythrin. Wells were incubated and washed and then mixed with 150 μL of sheath fluid to resuspend the beads. The plate was run on a Bio-Plex MAGPIX™ Multiplex Reader with Bio-Plex Manager™ MP Software (Luminex, Austin, TX, USA) and data were acquired. Cardiovascular markers were expressed as ng per mL of plasma.

#### 2.8.2. Cytokines and Chemokines

The determination of inflammatory factors was performed using the ProcartaPlex™ Mouse Cytokine & Chemokine Convenience Panel 1 26-Plex (ThermoFisher Scientific, Waltham, MA, USA), following manufacturer’s instructions. A panel of 26 cytokines and chemokines was measured for this study using a 96-well plate: (i) Th1/Th2 cytokines: GM-CSF, IFNɣ, IL-1β, IL-2, IL-4, IL-5, IL-6, IL-12p70, IL-13, IL-18, TNFα; (ii) Th9/Th17/Th22/Treg cytokines: IL-9, IL-10, IL-17A (CTLA-8), IL-22, IL-23, IL-27; (iii) chemokines: Eotaxin (CCL11), GROα (CXCL1), IP-10 (CXCL10), MCP-1 (CCL2), MCP-3 (CCL7), MIP-1α (CCL3), MIP-1β (CCL4), MIP-2 and RANTES (CCL5). Briefly, 25 mL of non-diluted sample were mixed with an assay buffer and capture beads in a 96-well plate using duplicates. The plate was incubated with shaking at room temperature for 2 h to facilitate the reaction. After incubation, 25 mL of detection antibodies were added, and the plate was incubated with shaking at room temperature for 30 min. Following incubation, 50 μL of Streptavidin-PE were added to each well. After incubating and washing, 120 μL of reading buffer were added to wells, and the plate was incubated at room temperature for 5 min. Finally, the plate was run on a Bio-Plex MAGPIX™ Multiplex Reader with Bio-Plex Manager™ MP Software (Luminex, Austin, TX, USA), and data were acquired. Inflammatory markers were expressed as pg per mL of plasma.

#### 2.8.3. Protein Carbonyls and Pentosidine

The AAS determination was carried out as previously described [42]. Briefly, 50 µL of plasma were treated with cold 10% trichloroacetic acid (TCA) solution, vortexed and centrifuged at 600× *g* for 5 min at 4 °C. The supernatants were removed, and the pellets were incubated with freshly prepared solution composed of 0.5 mL of 250 mM 2-(*N*-morpholino) ethanesulfonic acid (MES) buffer, pH 6.0, containing 1 mM diethylenetriaminepentaacetic acid (DTPA); 0.5 mL of 50 mM 4-amino benzoic acid (ABA) in 250 mM of MES buffer pH 6.0 and 0.25 mL 100 mM sodium cyanoborohydride (NaBH_3_CN) in 250 mM MES buffer pH 6.0. Samples were vortexed and incubated at 37 °C for 90 min, stirring them every 15 min. After derivatization, samples were treated with cold 50% TCA solution and centrifuged at 1200× *g* for 10 min. Supernatants were removed and pellets were washed twice with 10% TCA and diethyl ether-ethanol (1:1). Finally, pellets were treated with 6N HCl and incubated at 110 °C for 18 h until completion of hydrolysis. The hydrolysates were dried in vacuo in a centrifugal evaporator. The generated residues were reconstituted with 200 µL of Milli-Q water and filtered through hydrophilic polypropylene GH Polypro (GHP) syringe filters with 0.45 μm pore size (Pall Corporation, Port Washington, NY, USA) for HPLC analysis. Samples were analyzed using a Shimadzu Prominence HPLC instrument equipped with a quaternary solvent delivery system (LC-20AD), DGU-20AS on-line degasser, SIL-20A auto-sampler, RF-10A XL fluorescence detector (FLD) and CBM-20A system controller (Shimadzu Corporation, Kyoto, Japan). An aliquot (1 μL) from the reconstituted protein was injected for the analysis. AAS-ABA, GGS-ABA and pentosidine were eluted in a Cosmosil 5C_18_-AR-II RP-HPLC column (150 × 4.6 mm × 5 µm) equipped with a guard column (10 × 4.6 mm). The flow rate was kept at 1 mL min^−1^, and the temperature of the column was maintained constant at 30 °C. The eluate was monitored with excitation and emission wavelengths set at 283 and 350 nm, respectively. Standards (0.1 μL) were run and analyzed under the same conditions. Identification of both derivatized semialdehydes in the fluorescence detector chromatograms was carried out by comparing their retention times with those from the standard compounds. The peaks corresponding to analytes of interest (AAS-ABA, GGS-ABA and pentosidine) were manually integrated from fluorescence detector chromatograms and the resulting areas from derivatized semialdehydes plotted against an ABA standard curve with known concentrations, that ranged from 0.1 to 0.5 mM. Results were expressed as nmol of semialdehyde AAS per mg of protein. Pentosidine was not quantified and, hence, expressed as fluorescence intensity.

### 2.9. Bioinformatics and Statistical Analysis

Data in the graphs are expressed as bacterial composition (percentages), mean and standard error of the mean (mean ± SEM), and median and interquartile range. All determinations in mouse feces and plasmas were performed in five groups with six to eight animals per group (*n* = 6–8).

Regarding the bacterial diversity, the Chao1 and Shannon indices (i.e., alpha diversity) were used to measure the number and diversity of bacteria in the microbial community, and the Bray-Curtis dissimilarity index (i.e., beta diversity) was used to measure the dissimilarity in the microbial community composition of each group based on abundance data. Statistically, alpha diversity was assessed using the Kruskal-Wallis rank-sum test and the Benjamini-Hochberg procedure for multiple comparisons, and beta diversity was assessed using the analysis of similarity (ANOSIM). Principal coordinate analysis (PCoA) based on Bray-Curtis dissimilarity metrics was used to show the distance in the bacterial communities between the treatment groups.

The distribution of raw data from biochemical determinations was assessed using the D’Agostino-Pearson normality test in order to use parametric or non-parametric tests. Differences in normal variables were assessed using the Student’s *t* test (two groups) or one-way analysis of variance (ANOVA) (more than two groups). In contrast, non-normal variables were assessed using the Mann-Whitney U test (two groups) or the Kruskal-Wallis rank-sum test (more than two groups) as non-parametric tests. When raw data were not normally distributed because of a positively skewed distribution, data were log_10_-transformed to approximate a normal distribution and to ensure statistical assumptions of the parametric tests. The Sidak’s correction test was used as post hoc tests for multiple comparisons in the ANOVA and Kruskal-Wallis rank-sum test.

Correlation analyses were performed to examine the association between biochemical and/or microbial variables using the Spearman’s correlation coefficient (rho) with categorical variables and the Pearson’s correlation coefficient (r) with continuous variables.

Quantitative Insights into Microbial Ecology (QIIME2, version 2019.4) software [43] was used to analyze sequence quality and for diversity and taxonomic analysis, as previously described [44]. The statistical analysis of the microbiota sequencing was performed in R version 3.6.0., while the statistical analysis of biochemical data was performed using the Graph-Pad Prism version 5.04 software (GraphPad Software, San Diego, CA, USA). Test statistic values and degrees of freedom are indicated in the results. A *p*-value less than 0.05 was considered statistically significant. 

## 3. Results

### 3.1. Mice from the Experimental Groups Showed Differences in the Composition and Bacterial Abundances of Gut Microbiota

To characterize and identify intestinal bacteria populations in the fecal samples of mice from the experimental groups, we assessed diversity indices and examined the gut microbiota at different taxa levels.

#### 3.1.1. Alpha and Beta Diversity

Regarding the alpha diversity, the Chao1 (Figure 1A) and Shannon (Figure 1B) indices at genus level showed no significant differences in bacterial richness and diversity of samples from mice of the experimental groups (control, carnitine, savory, parsley and rosemary groups). In contrast, the Bray-Curtis dissimilarity index showed a significant dissimilarity in the microbial community composition of fecal samples comparing the mean of ranked dissimilarities between and within groups (R = 0.175, *p* = 0.001, ANOSIM) (Figure 1C). The PCoA of the Bray-Curtis distance matrix based at genus level abundances allowed for representation of the dissimilarities of each sample and showed the percentage of variation explained by the principal coordinates (PCoA1 = 32% and PCoA2 = 19%) (Figure 1D).

#### 3.1.2. Gut Microbial Abundances at Different Taxonomic Levels

The relative abundances at the phylum level in the experimental groups showed Bacteroidetes, Firmicutes and Proteobacteria as the most predominant phyla, but there were no differences among the groups (Figure 2A). In contrast, variations in bacterial communities were observed among the different groups at the family and genus levels. Thus, while there were no differences in the relative abundance between the carnitine and control groups, the essential oil groups showed relevant differences in bacterial families and genera. At the family level, there was a higher abundance of Lactobacillaceae and a lower abundance of Bacteroidaceae in mice treated with essential oils (mainly in the parsley and rosemary groups) than in mice from the carnitine group (Figure 2B). Similarly, mice treated with essential oils had a higher abundance of *Lactobacillus* and a lower abundance of *Bacteroides* than mice from the carnitine group (Figure 3A). Furthermore, the analysis of the number of each genus revealed significant differences in *Lactobacillus* genus among the groups (*p* < 0.040), and mice treated with essential oils, mainly in the rosemary group, had a higher number of *Lactobacillus* than mice from the carnitine and control groups (Figure 3B). The relative abundances at the species level are shown in Appendix A.

### 3.2. Treatment with Essential Oils of Parsley and Rosemary Reduced Plasma TMAO Levels

L-carnitine, TMA and TMAO levels were measured in the plasma of mice from all groups after treatment. Raw data were log10-transformed, and the estimated marginal means and SEM of the logarithmic values are represented in Figure 4.

#### 3.2.1. Plasma L-carnitine Levels

As expected, L-carnitine supplementation induced a significant increase in plasma L-carnitine levels in the carnitine group compared with the control group (t_7.194_ = 5.269, *p* = 0.001). However, one-way ANOVA revealed no significant differences in the L-carnitine levels in the essential oil (i.e., savory, parsley and rosemary) and carnitine groups (Figure 4A).

#### 3.2.2. Plasma Levels of TMA and TMAO

Similar to L-carnitine levels, mice from the carnitine group showed a significant increase in plasma TMA levels compared with the control group (t_11_ = 3.706, *p* = 0.004), but there were no significant differences in TMA levels in the essential oil and carnitine groups (Figure 4B).

Unlike L-carnitine and TMA levels, there were no significant differences in plasma TMAO levels between the carnitine and control groups. However, there were significant differences in TMAO levels among the essential oil and carnitine groups (F_3,23_ = 4.890, *p* = 0.009), and the post hoc test showed significant decreases in the parsley and rosemary groups compared with the carnitine group (*p* < 0.05) (Figure 4C).

### 3.3. Treatment with Essential Oils of Parsley and Rosemary Increased Fecal SCFAs Levels

Acetic, butyric and propionic acids were measured in the feces of mice from the experimental groups, as shown in Figure 5.

The comparisons of these SCFA species between the carnitine and control groups revealed no significant differences. In contrast, one-way ANOVA tests showed significant differences in fecal levels of acetic (F_3,25_ = 3.624, *p* = 0.028), butyric (F_3,25_ = 14.07, *p* < 0.001) and propionic (F_3,24_ = 4.093, *p* = 0.018) acids among mice from the essential oil and carnitine groups. Specifically, the post hoc comparison showed significant increases in acetic acid levels in the parsley and rosemary groups compared with the carnitine group (*p* < 0.05) (Figure 5A); significant increases in propionic acid levels in the parsley and rosemary groups compared with the carnitine group (*p* < 0.05 and *p* < 0.001, respectively) (Figure 5B); and a significant increase in butyric acid levels in the rosemary group compared with the carnitine group (*p* < 0.05) (Figure 5C).

### 3.4. Treatment with Essential Oils of Savory and Parsley Increased Plasma Thrombomodulin Levels

Plasma samples were used to measure common cardiovascular disease markers (sE-selectin, sICAM-1, Precam-1, sP-selectin, PAI-1 and thrombomodulin). Raw data were log10-transformed, and the estimated marginal means and SEM of the logarithmic values are represented in Figure 6.

The comparisons of these cardiovascular disease markers between the carnitine and control groups revealed no significant differences. The one-way ANOVA tests showed no significant differences in sE-selectin (Figure 6A), sICAM-1 (Figure 6B), Precam-1 (Figure 6C), sP-selectin (Figure 6D) and PAI-1 (Figure 6E) levels among mice from the essential oil and carnitine groups. In contrast, there were significant differences in thrombomodulin levels (F_3,22_ = 8.523, *p* < 0.001), and the post hoc comparisons showed significant increases in the savory and parsley groups compared with the carnitine group (*p* < 0.001 and *p* < 0.01, respectively) (Figure 6F).

### 3.5. Treatment with Essential Oils Altered Plasma Inflammatory Markers

Cytokines and chemokines were analyzed in the plasma of mice from the experimental groups. Because of statistical requirements, raw data were log10−transformed, and the estimated marginal means and SEM were represented and statistically analyzed. For clarity, only those cytokines and chemokines that resulted in statistical significance were shown in Figure 7.

#### 3.5.1. Cytokines

The comparisons between mice from the carnitine and control groups only showed differences in IFNɣ levels (Figure 7A), and a significant increase was found in the carnitine group compared with the control group (t_6.571_ = 2.480, *p* = 0.044).

Regarding one-way ANOVAs and post hoc tests among the essential oil and carnitine groups, there were significant differences in the plasma levels of various cytokines: (a) IFNɣ (F_3,23_ = 3.690, *p* = 0.027), the parsley and rosemary groups showed significant decreases compared with the carnitine group (*p* < 0.05) (Figure 8A); (b) TNFα (F_3,23_ = 4.923, *p* = 0.009), the rosemary group showed a significant decrease compared with the carnitine group (*p* < 0.05) (Figure 7B); IL-4 (F_3,23_ = 3.876, *p* = 0.022), the savory group showed a significant decrease compared with the carnitine group (*p* < 0.05) (Figure 7C); IL-6 (F_3,23_ = 3.200, *p* = 0.042), the savory group showed a significant increase compared with the carnitine group (*p* < 0.05) (Figure 7D); IL-12p70 (F_3,23_ = 5.090, *p* = 0.008), the rosemary showed a significant decrease compared with the carnitine group (*p* < 0.01) (Figure 7E); IL-22 (F_3,23_ = 5.129, *p* = 0.007), the savory, parsley and rosemary groups showed significant decreases compared with the carnitine group (*p* < 0.01, *p* < 0.05 and *p* < 0.05, respectively) (Figure 7F); and IL-23 (F_3,23_ = 3.302, *p* = 0.038), the savory group showed a significant increase compared with the carnitine group (*p* < 0.05) (Figure 7G).

Other cytokines (i.e., GM-CSF, IL-1β, IL-2, IL-5, IL-9, IL-10, IL-13, IL-17A, IL-18 and IL-27) were not significantly altered in the experimental groups (Appendix A).

#### 3.5.2. Chemokines

The comparisons between the carnitine and control groups only showed differences in CXCL10 levels (Figure 7H), and a significant increase was observed in the carnitine group compared with the control group (t_12_ = 2.664, *p* = 0.037). However, there were no significant differences in the essential oil and carnitine groups.

Unlike CXCL10, one-way ANOVA tests revealed significant differences in other chemokines as follows: (a) CXCL1 (F_3,23_ = 5.466, *p* = 0.006), the savory groups showed a significant increase compared with the carnitine group (*p* < 0.01) (Figure 7I); (b) CCL2 (F_3,23_ = 4.597, *p* = 0.012), the savory groups showed a significant increase compared with the carnitine group (*p* < 0.01) (Figure 7J); and CCL11 (F_3,23_ = 3.109, *p* = 0.046), the savory groups showed a significant increase compared with the carnitine group (*p* < 0.05) (Figure 7K). Therefore, only the savory group showed significant increases in chemokines.

Similar to cytokines, other chemokines (i.e., CXCL2, CCL3, CCL4, CCL5 and CCL7) were not significantly altered in the experimental groups (Appendix A).

### 3.6. Treatment with Essential Oils Reduced Protein Oxidative Stress 

Protein carbonyls (AAS and GGS) and pentosidine were also measured in the plasma of mice from the experimental groups to examine the effects of essential oil emulsions on oxidative stress, as shown in Figure 8.

#### 3.6.1. Protein Carbonyls

The comparisons between the carnitine and control groups revealed that supplementation with L-carnitine had no significant effects on the plasma levels of AAS and GGS. However, one-way ANOVA tests revealed significant differences in AAS (F_3,24_ = 4.238, *p* = 0.015) (Figure 8A) and GGS (F_3,24_ = 3.421, *p* = 0.033) (Figure 8B) in the essential oil and carnitine groups. Thus, the post hoc comparisons showed a significant decrease in AAS levels in the savory group and a significant decrease in GGS levels in the rosemary group compared with the carnitine group (*p* < 0.05).

#### 3.6.2. Pentosidine

Although there were no differences between the carnitine and control groups, significant differences in pentosidine levels were found in the essential oil and carnitine groups (F_3,24_ = 11.09, *p* < 0.001) (Figure 8C). Similar to AAS, the savory group showed a significant decrease in pentosidine levels compared with the carnitine (*p* < 0.001).

### 3.7. Association between Gut Microbial Abundance and Metabolites

Because the TMAO precursor (TMA) and SCFA species are produced by gut microbiota from dietary fiber fermentation and carnitine metabolism, we investigated the association between these products and the microbial abundance at different taxonomic ranks. Additionally, we explored the association between inflammatory and oxidative markers and the microbial abundance.

#### 3.7.1. L-carnitine, TMA and TMAO

As shown in Table 1, we analyzed the association between the abundance of gut microbiota and plasma levels of L-carnitine, TMA and TMAO. The analysis of these metabolites revealed significant and positive correlations between L-carnitine and TMA levels (r = +0.77, *p* < 0.001) and between TMA and TMAO levels (r = +0.49, *p* < 0.01). Regarding the microbial abundance, significant associations with L-carnitine and TMA levels were found at different taxonomic ranks. At the phylum level, L-carnitine levels were inversely correlated with Tenericutes (rho = −0.36, *p* < 0.05) and TMA levels were positively correlated with Lentisphaerae (rho = +0.34, *p* < 0.05), but inversely correlated with Tenericutes (rho = −0.37, *p* < 0.05). At the family level, L-carnitine levels were inversely correlated with Anaeroplasmataceae (rho = −0.36, *p* < 0.05) and TMA levels were positively correlated with Lactobacillaceae (rho = +0.35, *p* < 0.05), Alcaligenaceae (rho = +0.34, *p* < 0.05) and Victivallaceae (rho = +0.34, *p* < 0.05), but inversely correlated with Anaeroplasmataceae (rho = −0.37, *p* < 0.05). At the genus level, L-carnitine levels were inversely correlated with *Anaeroplasma* (rho = −0.36, *p* < 0.05) and TMA levels were positively correlated with *Lactobacillus* (rho = +0.35, *p* < 0.05), *Pigmentiphaga* (rho = +0.39, *p* < 0.05) and *Victivallis* (rho = +0.34, *p* < 0.05), but inversely correlated with *Anaeroplasma* (rho = −0.37, *p* < 0.05). In contrast, TMAO levels were not significantly associated with the abundance of gut microbiota.

#### 3.7.2. SCFA Species

The association between the abundance of gut microbiota and fecal levels of acetic, propionic and butyric acids was also analyzed (Table 2). A first correlation analysis between SCFA levels showed significant and positive correlations (acetic acid vs. propionic acid, r = +0.77, *p* < 0.001; acetic acid vs. butyric acid, r = +0.69, *p* < 0.001; and propionic acid vs. butyric acid, r = +0.66, *p* < 0.001). At the family level, acetic acid levels were positively correlated with Lactobacillaceae (rho = +0.39, *p* < 0.05) and propionic acid levels were positively correlated with Barnesiellaceae (rho = +0.40, *p* < 0.05), Odoribacteraceae (rho = +0.38, *p* < 0.05) and Lactobacillaceae (rho = +0.36, *p* < 0.05). At the genus level, we confirm these significant associations with acetic (*Lactobacillus*) and propionic (*Barnesiella*, *Butyricimonas* and *Lactobacillus*) acids. Unlike acetic and propionic acids, there were no significant correlations between the gut microbial abundance and butyric acid levels.

#### 3.7.3. Cytokines and Chemokines

Additionally, we also performed a correlation analysis to explore the association between gut microbiota and inflammatory markers (Appendix A). After adjustment for multiple correlations, we found significant associations with some inflammatory markers at family and genus levels. At the family level, there were inverse correlations between IL−1ß levels and Oxalobacteraceae (rho = −0.46, adjusted *p* < 0.05), GM-CSF levels and Lactobacillaceae (rho = −0.53, adjusted *p* < 0.05), and CXCL12 levels and Ruminococcaceae (rho = −0.46, adjusted *p* < 0.05). At the genus level, there were inverse correlations between GM-CSF levels and *Lactobacillus* (rho = −0.53, adjusted *p* < 0.05), CXCL12 levels and *Ocillospira* (rho = −0.46, adjusted *p* < 0.05), CXCL12 levels and *Bilophila* (rho = −0.46, adjusted *p* < 0.05), and CCL7 levels and *Ocillospira* (rho = −0.49, adjusted *p* < 0.05); but there was a positive correlation between IL−22 levels and *Clostridium* (rho = +0.47, adjusted *p* < 0.05).

#### 3.7.4. AAS, GGS and Pentosidine

Overall, there was a negative association between the gut microbial abundance and plasma levels of carbonyl products and pentosidine (Appendix A). After adjustment for multiple correlations, we found significant correlations with AAS and GGS levels at family and genus levels. Namely, AAS levels were inversely correlated with Paraprevotellaceae and *Paraprevotella* (rho = −0–47, adjusted *p* < 0.05), GGS levels were inversely correlated with Barnesiellaceae and *Barnesiella* (rho = −0.56, adjusted *p* < 0.05) and Lactobacillaceae and *Lactobacillus* (rho = −0.50, adjusted *p* < 0.05). Unlike AAS and GGS, pentosidine levels were not significantly associated with the gut microbial abundance.

## 4. Discussion

Evidence supports that adherence to the Mediterranean diet is strongly associated with a reduction in the risk of cardiovascular disease [45,46,47]. In fact, the Mediterranean diet is recommended to patients, as it was shown to be effective for prevention of cardiovascular events [48,49]. Some components of the Mediterranean diet such as olive oil provide cardiovascular benefits, antithrombotic properties [50,51] and improves postprandial lipemia concentration, which is typically elevated in patients with T2DM [52]. In recent years, interest in functional components from herbal medicines has increased, supported by the confirmed medicinal potential of essential oils [53]. In this study, we aimed to assess the effects of essential oils from savory, parsley and rosemary, which are commonly used condiments in the Mediterranean diet, using a higher dose than within a regular diet. We used the essential oils as potential nutraceuticals and assessed their effects on microbial populations, their metabolites (TMA and TMAO in plasma and SCFAs in feces) and plasma markers (cardiovascular disease, inflammation and oxidative stress), using a humanized mouse model harboring colonic microbiota derived from that of patients with IHD and T2DM. The main results of this study are as follows: (a) Treatments with essential oil emulsions of savory, parsley and rosemary had prebiotic effects on gut microbiota by inducing an increase in *Lactobacillus* genus, which are considered beneficial bacteria; (b) Plasma TMAO levels, a pro-atherogenic substance related to the pathogenicity of IHD and the production of pro-inflammatory cytokines [54], were significantly reduced after treatment with essential oils, more specifically with parsley and rosemary; (c) Fecal levels of SCFA species were increased after treatment with parsley and rosemary essential oils, which suggests a beneficial effect of these essential oils on the gastrointestinal health; (d) Plasma thrombomodulin levels were increased after treatments with essential oils of savory and parsley; (e) Overall, essential oils had anti-inflammatory effects through alterations in the plasma levels of cytokines and chemokines; and (f) Finally, there was a reduction in the expression of protein carbonyls and pentosidine.

High-level adherence to the Mediterranean diet has been positively associated with changes in gut microbiota composition and their metabolites [55]. Gut microbiota uses L-carnitine as a precursor to generate TMA, which is rapidly absorbed into the portal circulation by passive diffusion across the enterocyte membranes and then oxidized to TMAO by the action of hepatic flavin-containing monooxygenases (mainly FMO3) [56]. As expected, our results showed that TMA was only increased after administration of L-carnitine, but we did not observe changes in TMA levels after treatments with different essential oil emulsions. In contrast, we observed a significant reduction of plasma TMAO levels in the parsley and rosemary groups that cannot be explained by changes in the TMA levels. Therefore, the treatments with parsley and rosemary essential oils likely affect the metabolism of TMAO by decreasing the oxidative activity of FMO3 and/or increasing the mobilization of TMAO (i.e., absorption by tissues or excretion in urine) [57]. However, the exact mechanism has to be elucidated, and further research is needed.

Among others, the microorganisms linked to high concentrations of pro-atherogenic substances, mainly TMAO, include species from Firmicutes, Pseudomonas, Bacillota and Proteobacteria phyla, such as *Anaerococcus hydrogenalis*, *Clostridium asparagiforme*, *Clostridium hathewayi*, *Clostridium sporogenes*, *Escherichia fergusonii*, *Proteus penneri* or *Providencia rettgeri* [58]. In fact, these microorganisms were found to a lesser extent in the colonic microbiota of patients presenting low levels of TMAO in plasma [59]. However, after correlation analysis, we found that TMAO levels were not significantly associated with the abundance of different groups of gut microbiota. This fact would support the idea that the TMAO inhibition provoked by these essential oils would be based on the TMAO oxidation hampering, instead of the bacterial role.

Specific dietary components could also alter gut microbiota composition and activity [60]. Colonic microbiota populations play a key role in the generation of SCFAs, with a positive impact on the metabolism of the host [16]. Within the SCFA, acetate is present in highest proportions in subjects with adherence to the Mediterranean diet [60]. In our results, we observed that Mediterranean plant-derived essential oils alter the fecal composition of SCFA species, as previously reported [61]. More specifically, essential oil emulsions of parsley and rosemary induced an increase in the acetic acid levels, the most abundant SCFA. These results are consistent with the increase in *Lactobacillus* genus population observed in our results, which is one of the highest contributors to the production of acetic acid [62]. In addition, we also observed an increase in the fecal levels of propionic and butyric acids in the rosemary group. In this regard, the higher content of SCFAs seems to be also linked to the increased percentage of *Barnesiella* genus in the gut microbiota by the effect of rosemary essential oil. Consistently, the positive correlation of this genus with the SCFA levels [63] has been previously reported.

Previous studies demonstrated that SCFAs have a beneficial effect on regulating regulatory T cells [64], revealing the important role of both microbiota and microbiota-derived SCFAs on immune system modulation [65,66]. Our data showed a marked differential effect on plasma inflammatory markers depending on the essential oil administered. Thus, our results suggest a role of savory essential oil in a pro-inflammatory response, while both parsley and rosemary essential oils induce an anti-inflammatory profile, likely linked to a high production of SFCAs. It is worth emphasizing that not all SFCAs play the same role in relation to gut health, lipid metabolism and health status. Acetic and butyric acids are used by rat colonic epithelial cells as an energy source and strengthen the epithelial homeostasis preventing inflammation [67,68]. Considering the observed prebiotic effect, boosting *Lactobacillus* genus among others, in parallel to the higher levels found for the three main SFCAs and the anti-inflammatory response, rosemary essential oil seems to exert the most beneficial effect.

In previous studies, the Mediterranean diet has been associated with reduced inflammation [69] due to the effect of some of its components. In our study, the analysis of plasma inflammatory markers, cytokines and chemokines, showed different profiles depending on the type of essential oil used in the treatment. Thus, savory essential oil induced an increase in the levels of chemokines CXCL1, CCL2 and CCL11 and pro-inflammatory cytokines IL−6 and IL−23. Moreover, savory essential oil induced a decrease in IL−4 levels, which is a typical anti-inflammatory cytokine. On the contrary, parsley essential oil induced a more anti-inflammatory pattern showing low levels of IFNγ and IL−22 cytokines. Finally, treatment with rosemary essential oil emulsion clearly showed an anti-inflammatory profile, reducing the levels of IFNγ, TNFα, IL−12p and IL−22. In addition, we found increased levels of CXCL10 in all groups supplemented with L-carnitine when compared with the control group. CXCL10 is a pro-inflammatory chemokine that can be secreted by numerous cell types in response to an inflammatory process, regulating cell recruitment [70]. Its function can be regulated by cytokines, such as IFNγ and TNFα [71,72,73], and it has been proposed, together with CXCL9 and CXCL11, as a biomarker for heart failure and left ventricular dysfunction [74,75]. It has been previously reported that essential oils can modulate the secretion of important cytokines, having an effect in inflammatory pathways such as nuclear factor kappa-light-chain-enhancer of activated B cells (NF-kB) [76]. Our results suggest that treatment with parsley and rosemary essential oils may partially compensate for the elevation of the pro-inflammatory chemokine CXCL10 by inducing an anti-inflammatory profile.

These pro- and anti-inflammatory profiles may be also related to the production of thrombomodulin, an anti-coagulant cofactor. Interestingly, recent studies have shown that thrombomodulin exhibits anti-inflammatory effects by inhibiting leukocyte recruitment [77]. In addition, the prototypical pro-inflammatory NF-kB signaling pathway has been shown to down-regulate thrombomodulin expression [78]; however, the elevated thrombomodulin expression in the savory and parsley groups was only associated with the decrease in the expression of the pro-inflammatory cytokine IL−22, because the expression of other pro-inflammatory cytokines was inconsistent in both groups. 

The antioxidant effect of some plant-derived compounds has been previously applied against a series of chronic diseases, such as IHD and T2DM [30]. Recently, essential oils of some plants have shown to exhibit important antioxidant activity [79]. To evaluate the antioxidant potential of savory, parsley and rosemary essential oils, we assessed the plasma levels of protein carbonyls and pentosidine after treatment. Protein carbonyls are the result of oxidation of lysine, arginine and proline residues in proteins, mostly to AAS and GGS. A pentosidine is a glycosylation end-product formed by the cross-link of a pentose between arginine and lysine residues of proteins. Production of both carbonyls and pentosidine are induced by oxidative reactions and, therefore, their levels are considered as biomarkers of oxidative stress [80]. Protein oxidation and the accretion of protein carbonyls is a pathological hallmark of multiple chronic diseases, such as T2DM, inflammatory bowel diseases and neurodegenerative disorders, among others [81]. Scientific evidence reports the onset of carbonyl stress in hyperglycemic conditions leading to pancreatic failure, insulin resistance and onset of T2DM [31,82]. In addition, pentosidine is commonly used as an indicator of T2DM complications, such as hypertension and heart failure [83]. We did not observe changes induced by L-carnitine supplementation, despite being an antioxidant compound. However, we found a general reduction in the plasma levels of AAS, GGS and pentosidine levels after treatments with essential oil emulsions. Specifically, AAS and pentosidine levels were reduced in mice treated with savory essential oil, while GGS levels were reduced with rosemary essential oil treatment. Consistently, gut microbiota populations were found negatively correlated with AAS and GGS levels in the plasma. Reduction in inflammation in the parsley and rosemary groups may alleviate the oxidative stress, leading to lower levels of protein carbonyls and pentosidine. The savory group also showed reduced oxidative stress, pointing out that the essential oil may contain compounds with intrinsic antioxidant properties. Our results suggest an antioxidant profile of plant-derived compounds widely used in the Mediterranean diet, such as savory, parsley and rosemary essential oils, when used as nutraceuticals.

We are aware that there are some limitations to the findings reported in this study. First, sample size was low, and a higher sample size would allow us to consolidate our results. Second, we used a preclinical humanized model, which entails a number of limitations when transferring the findings directly to patients. Third, randomly cycling female mice were used because the variability in 30 categories of behavioral, morphological, physiological, and molecular traits is not higher than in male mice [84], and because female mice housed in groups do not fight [85]; however, we are aware that future studies need to incorporate females and males in equal numbers with explicit comparison of the two sexes and that the inclusion of the estrous cycle stage of female mice contributes to a better characterization in several of these biochemical variables. Finally, we only assessed a single high dose of essential oil emulsions of savory, parsley and rosemary to evaluate their potential to protect against cardiovascular diseases. Furthermore, the combination of these essential oils could give rise to a synergistic beneficial effect higher than the one observed by each essential oil separately. Thus, our results pave the way for future translational studies assessing the minimal dose with the maximum effect and the potential toxic effect.

## 5. Conclusions

In summary, this study demonstrates that dietary supplementation of essential oils from parsley, savory and rosemary exert prebiotic effects by promoting or restoring beneficial bacteria populations in the gut of humanized mice with fecal transplantation from patients with IHD and T2DM. These effects on gut microbiota caused a decrease in plasma TMAO levels and an increase in fecal SCFA levels. Interestingly, treatments with essential oil emulsions were associated with an anti-inflammatory and antioxidant profile. It is worth mentioning that rosemary essential oil was the most promising nutraceutical for the treatment and/or prevention of cardiovascular events. Moreover, this work displays novel evidence by which plant-derived essential oils, commonly used in the Mediterranean diet, promote health and protect against inflammation and oxidative stress typically observed in IHD and/or T2DM. Further studies are warranted to validate our results in humans with the aim to modulate the gut microbiota and enhance biochemical biomarkers in patients with IHD and T2DM.

## Figures and Tables

**Figure 1 nutrients-14-04650-f001:**
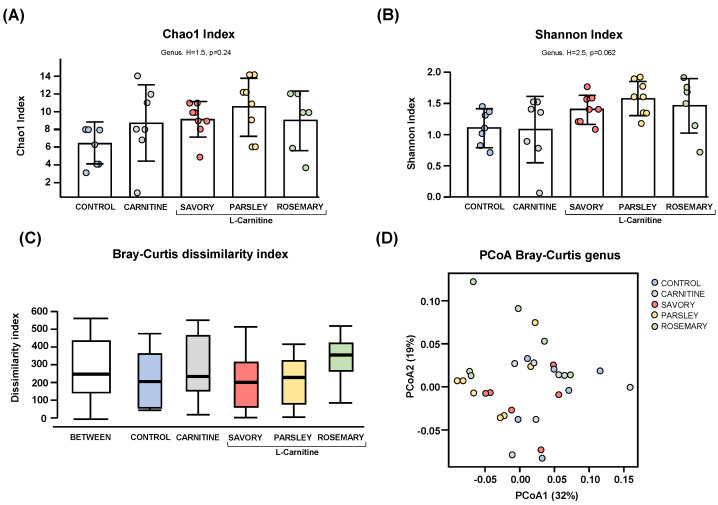
Alpha and beta diversity analyses of bacterial microbiota in fecal samples from the treatment groups. (**A**) Chaos1 index; (**B**) Shannon index; (**C**) Bray-Curtis dissimilarity index; and (**D**) Principal coordinates analysis of Bray-Curtis dissimilarity. Dots are individual values. Alpha diversity was assessed using the Kruskal-Wallis rank-sum test, and the Bray-Curtis dissimilarity index was assessed using the analysis of similarity (ANOSIM). Principal coordinates analysis (PCoA) based on Bray-Curtis dissimilarity index shows the distance in the bacterial communities between the treatment groups.

**Figure 2 nutrients-14-04650-f002:**
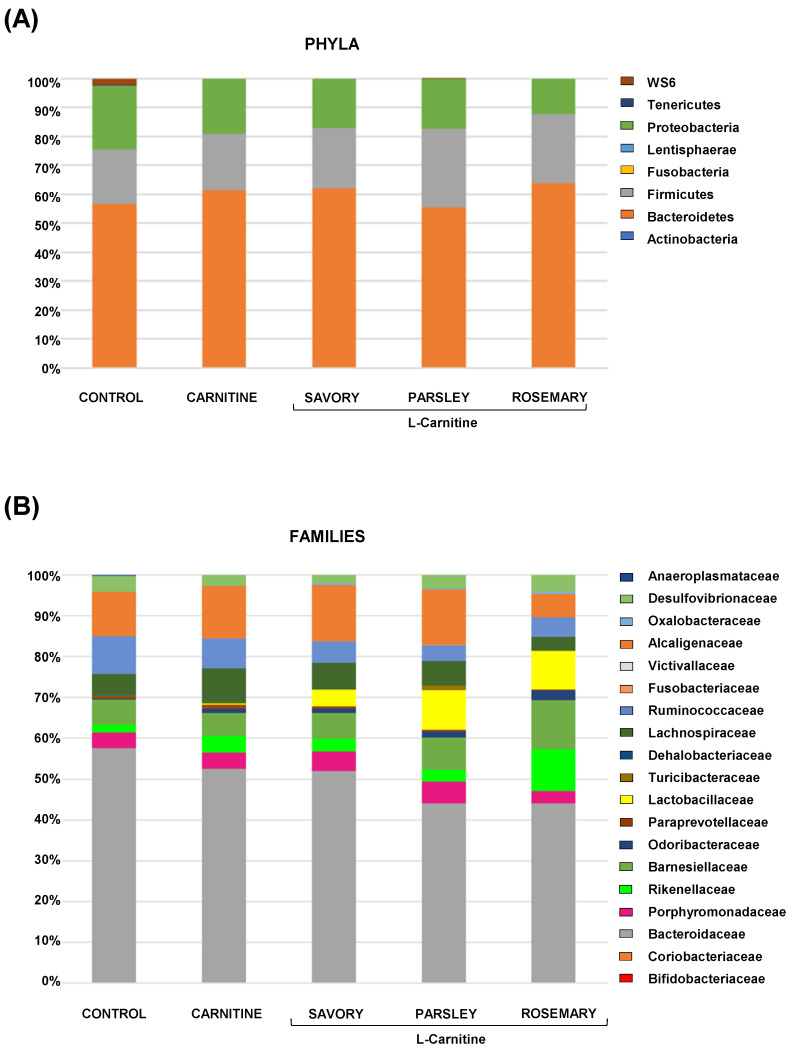
Bacterial profile at the phylum and family levels in fecal samples of mice from the treatment groups. (**A**) Phyla; and (**B**) Families. Bars show the relative abundances (%) for each group using 16S rRNA gene sequencing (Ion S5^TM^ System).

**Figure 3 nutrients-14-04650-f003:**
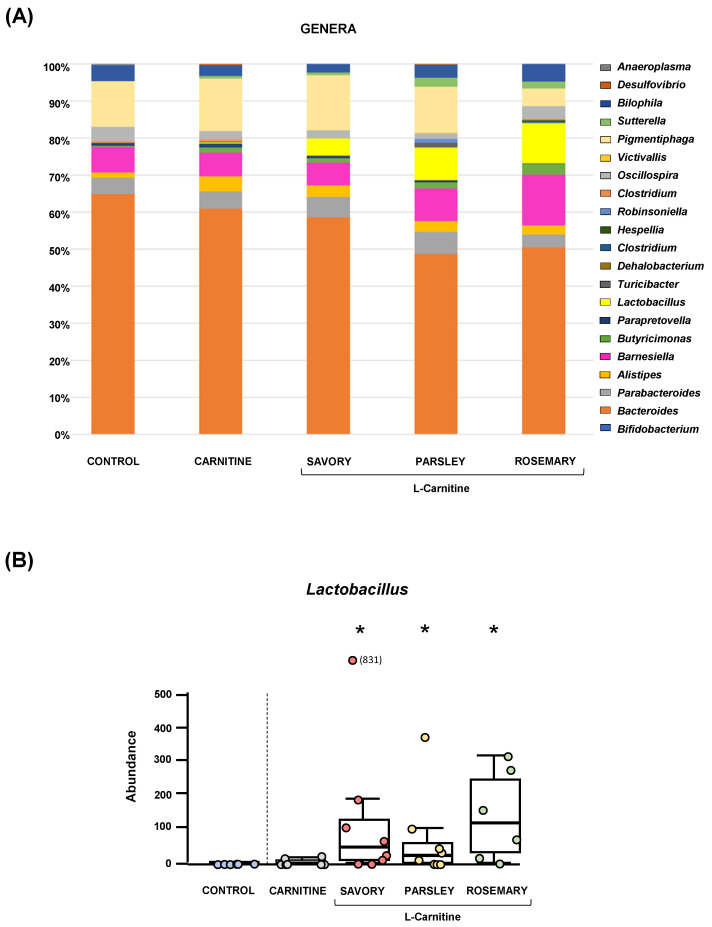
Bacterial profile at the genus level and abundance of Lactobacillus in fecal samples of mice from the treatment groups. (**A**) Genera; and (**B**) Lactobacillus. Bars show the relative abundance (%) for each group using 16S rRNA gene sequencing (Ion S5^TM^ System). Dots are individual values and data from the carnitine, savory, parsley and rosemary groups were analyzed using the Kruskal-Wallis rank-sum test (*) *p* < 0.05 denotes significant differences compared with the carnitine group.

**Figure 4 nutrients-14-04650-f004:**
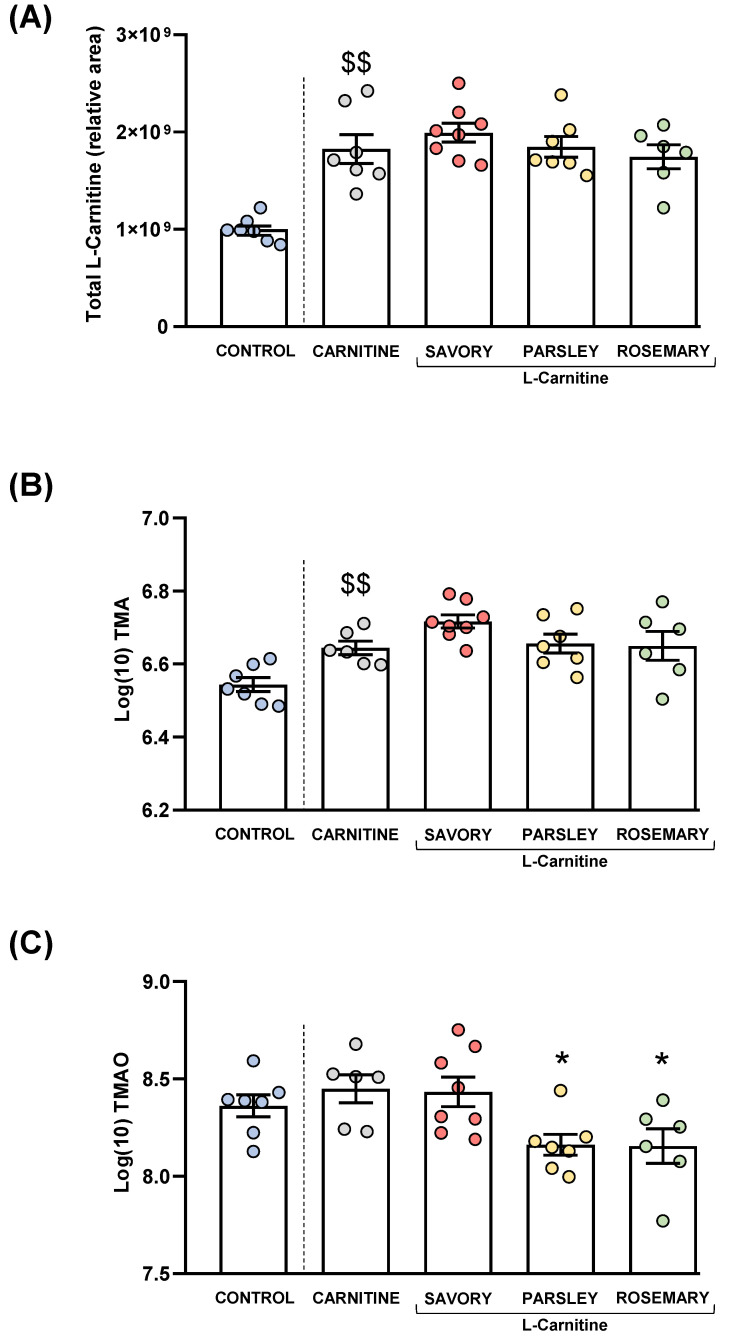
Plasma levels of total L-Carnitine, TMA and TMAO in mice from the treatment groups. (**A**) Total L-carnitine; (**B**) TMA; and (**C**) TMAO levels. Dots are individual values. Bars are means ± SEM of L-Carnitine concentrations (relative area) and log10-transformed concentrations of TMA and TMAO (ng mL^−1^). Data from the control and carnitine groups were analyzed using Student’s *t* test. Data from the carnitine, savory, parsley and rosemary groups were analyzed using one-way ANOVA. ($$) *p* < 0.01 denotes significant differences compared with the control group. (*) *p* < 0.05 denotes significant differences compared with the carnitine group.

**Figure 5 nutrients-14-04650-f005:**
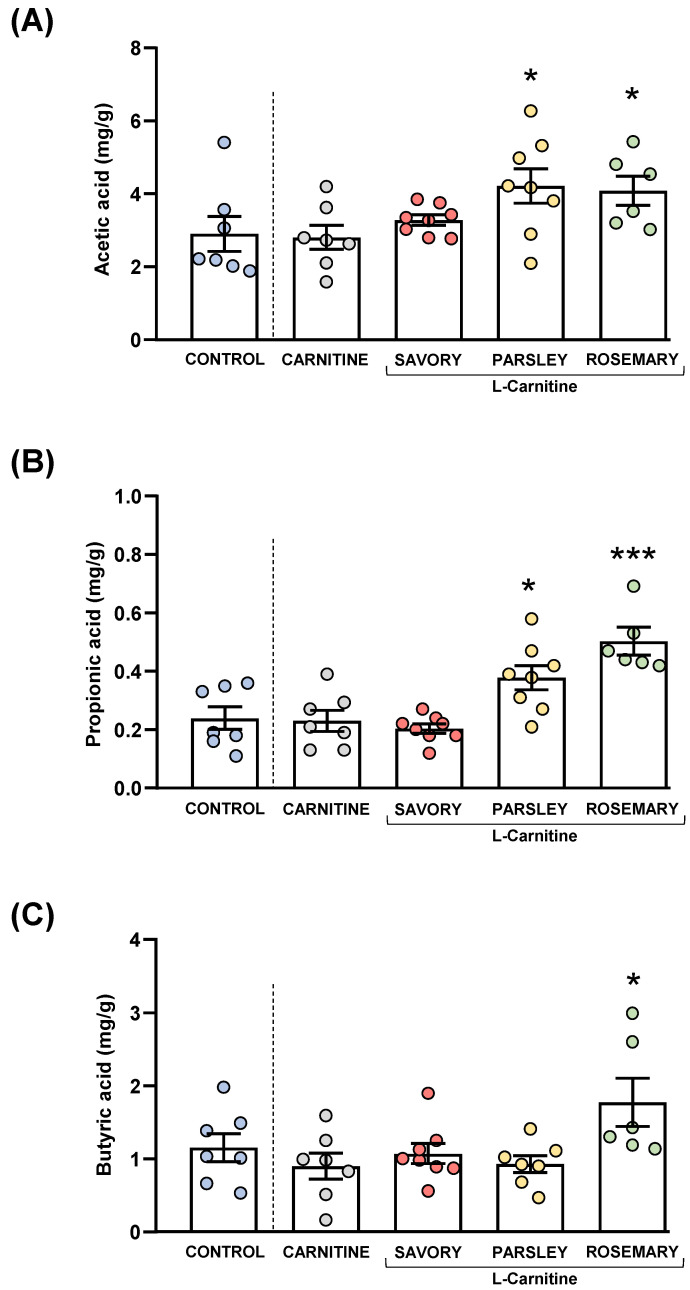
Fecal levels of SCFA species in mice from the treatment groups. (**A**) Acetic; (**B**) Propionic; and (**C**) Butyric acid levels. Dots are individual values. Bars are means ± SEM of SFCA concentrations (mg g^−1^). Data from the control and carnitine groups were analyzed using Student’s *t* test. Data from the carnitine, savory, parsley and rosemary groups were analyzed using one-way ANOVA. (*) *p* < 0.05 and (***) *p* < 0.001 denote significant differences compared with the carnitine group.

**Figure 6 nutrients-14-04650-f006:**
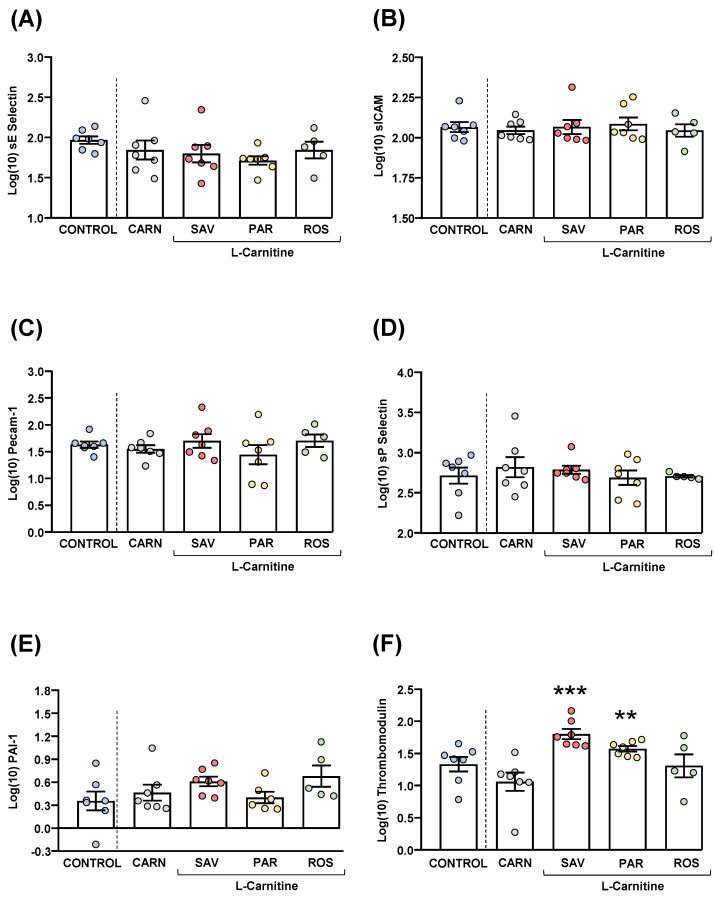
Plasma levels of cardiovascular markers in mice from the treatment groups. (**A**) sE-Selectin; (**B**) sICAM; (**C**) Pecam-1; (**D**) sP-Selectin; (**E**) PAI-1; and (**F**) Thrombomodulin levels. Dots are individual values. Bars are means ± SEM of log10-transformed concentrations of relevant cardiovascular markers (ng mL^−1^). Data from the control and carnitine groups were analyzed using Student’s *t* test. Data from the carnitine, savory, parsley and rosemary groups were analyzed using one-way ANOVA. (**) *p* < 0.01 and (***) *p* < 0.001 denote significant differences compared with the carnitine group.

**Figure 7 nutrients-14-04650-f007:**
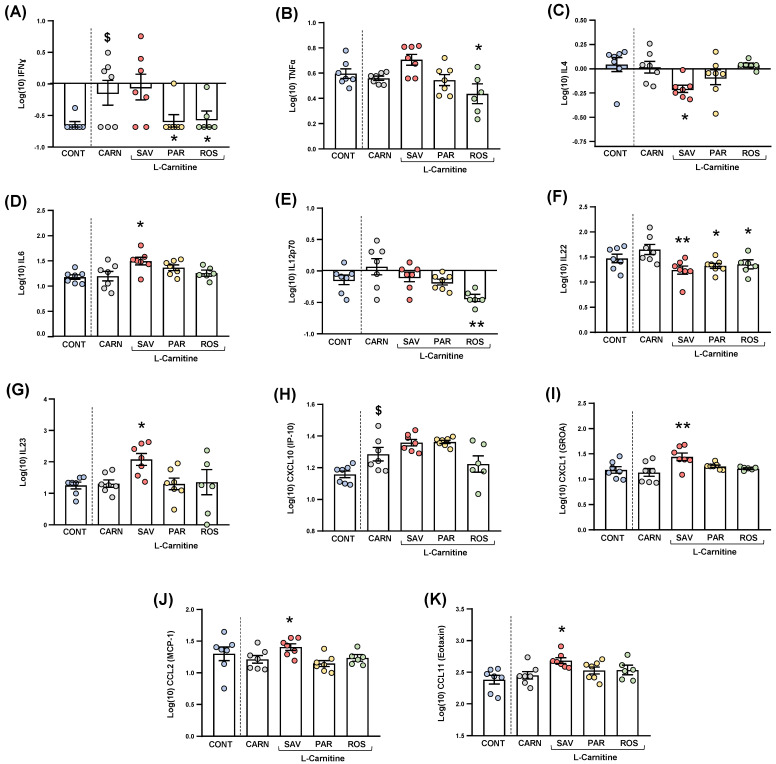
Plasma levels of cytokines and chemokines in mice from the treatment groups. (**A**) IFNɣ; (**B**) TNFα; (**C**) IL−4; (**D**) IL−6; (**E**) IL−12p70; (**F**) IL−22; (**G**) IL−23; (**H**) CXCL10 (IP−10); (**I**) CXCL1 (GROα); (**J**) CCL2 (MCP−1); and (**K**) CCL11 (Eotaxin) levels. Dots are individual values. Bars are means ± SEM of log10-transformed concentrations of cytokines and chemokines (pg mL^−1^). Data from the control and carnitine groups were analyzed using Student’s *t* test. Data from the carnitine, savory, parsley and rosemary groups were analyzed using one-way ANOVA. ($) *p* < 0.05 denotes significant differences compared with the control group. (*) *p* < 0.05 and (**) *p* < 0.01 denote significant differences compared with the carnitine group.

**Figure 8 nutrients-14-04650-f008:**
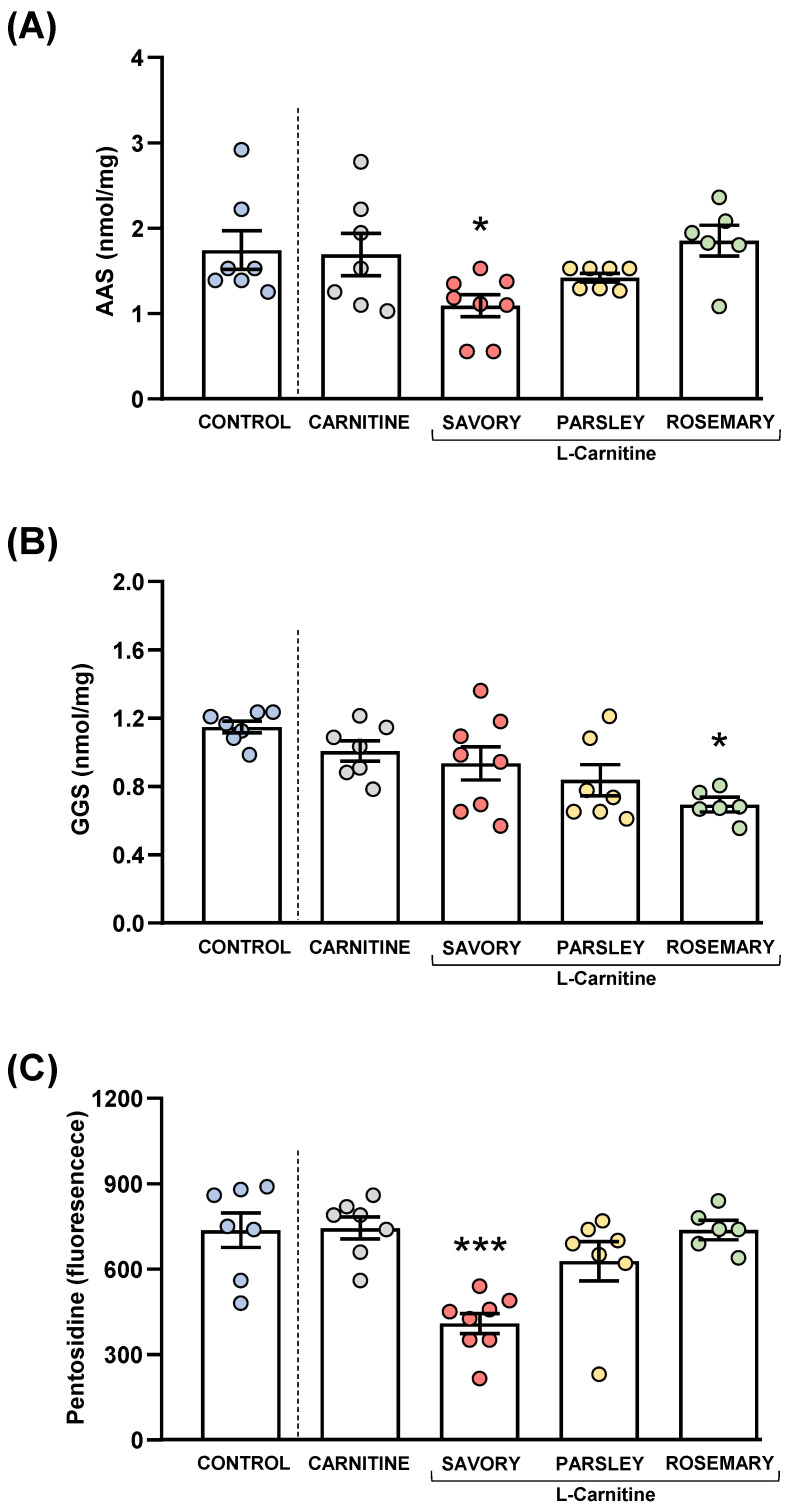
Plasma levels of protein carbonyls and pentosidine in mice from the treatment groups. (**A**) Aminoadipic semialdehyde (AAS); (**B**) Glutamic semialdehyde (GGS); and (**C**) Pentosidine levels. Dots are individual values. Bars are means ± SEM of AAS (nmol mg^−1^), GGS (nmol mg^−1^) and pentosidine (fluorescence intensity) concentrations. Data from the control and carnitine groups were analyzed using Student’s *t* test. Data from the carnitine, savory, parsley and rosemary groups were analyzed using one-way ANOVA. (*) *p* < 0.05 and (***) *p* < 0.001 denotes significant differences compared with the carnitine group.

**Table 1 nutrients-14-04650-t001:** Correlation analysis among plasma levels of L-carnitine, TMA and TMAO, as well as between gut microbial abundance and plasma levels of L-carnitine, TMA and TMAO.

	L-carnitine (Area)		TMA (ng mL^−1^)		TMAO (ng mL^−1^)	
	r	*p*-Value	r	*p*-Value	r	*p*-Value
TMAO (ng mL^−1^)	+0.152	0.383	+0.492	0.003	1	---
TMA (ng mL^−1^)	+0.767	<0.001	1	---	+0.492	0.003
L-carnitine (area)	1	---	+0.767	<0.001	+0.152	0.383
Phylum	rho	*p*-value	rho	*p*-value	rho	*p*-value
Lentisphaerae	+0.329	0.058	+0.342	0.047	+0.007	0.970
Tenericutes	−0.362	0.035	−0.369	0.032	+0.003	0.988
Family	rho	*p*-value	rho	*p*-value	rho	*p*-value
Lactobacillaceae	+0.337	0.051	+0.353	0.041	−0.210	0.234
Alcaligenaceae	+0.271	0.122	+0.342	0.048	+0.203	0.250
Victivallaceae	+0.329	0.058	+0.342	0.047	+0.007	0.970
Anaeroplasmataceae	−0.362	0.035	−0.369	0.032	+0.003	0.988
Genus	rho	*p*-value	rho	*p*-value	rho	*p*-value
*Lactobacillus*	+0.337	0.051	+0.353	0.041	−0.210	0.234
*Pigmentiphaga*	+0.248	0.157	+0.390	0.023	+0.259	0.139
*Victivallis*	+0.329	0.058	+0.342	0.047	+0.007	0.970
*Anaeroplasma*	−0.362	0.035	−0.369	0.032	+0.003	0.988

Abbreviations: r, Pearson correlation coefficient; rho, Spearman correlation coefficient; TMA, trimethylamine; TMAO, trimethylamine N−oxide.

**Table 2 nutrients-14-04650-t002:** Correlation analysis among fecal levels of SCFA species, as well as between gut microbial abundance and fecal levels of SCFA species.

	Acetic Acid (mg g^−1^)		Propionic Acid (mg g^−1^)		Butyric Acid (mg g^−1^)	
	r	*p*-Value	r	*p*-Value	r	*p*-Value
Acetic acid (mg g^−1^)	1	---	+0.770	<0.001	+0.692	<0.001
Propionic acid (mg g^−1^)	+0.770	<0.001	1	---	+0.655	<0.001
Butyric acid (mg g^−1^)	+0.692	<0.001	+0.655	<0.001	1	---
Family	rho	*p*-value	rho	*p*-value	rho	*p*-value
Barnesiellaceae	+0.233	0.177	+0.403	0.016	+0.162	0.352
Odoribacteraceae	+0.272	0.114	+0.375	0.027	+0.095	0.589
Lactobacillaceae	+0.387	0.022	+0.362	0.033	+0.054	0.758
Genus	rho	*p*-value	rho	*p*-value	rho	*p*-value
*Barnesiella*	+0.236	0.172	+0.411	0.014	+0.165	0.345
*Butyricimonas*	+0.272	0.114	+0.375	0.027	+0.095	0.589
*Lactobacillus*	+0.387	0.022	+0.362	0.033	+0.054	0.758

Abbreviations: r, Pearson correlation coefficient; rho, Spearman correlation coefficient; SCFA, short-chain fatty acid.

## Data Availability

Data is presented as individual data points in this study. Raw data is contained within the Appendix A. Additional information is available on request from the corresponding author.

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
