# Peer review of "Beneficial Effects of Essential Oils from the Mediterranean Diet on Gut Microbiota and Their Metabolites in Ischemic Heart Disease and Type-2 Diabetes Mellitus"

_nutrients, 2022, doi:10.3390/nu14214650_

Round 1

Reviewer 1 Report

In general, this is an essential and exciting topic that I enjoyed reading. Your team had tried a lot of experiments to provide evidence with a well-designed mice study. However, there is minor revision required with my concern.

1. References are required on page 3, lines 112-116.

2. Additional information is needed on page 4, lines 162-167. How did you free the stool samples from patients? Did you use glycerol to freeze the stool samples?? If you are not, you need to clarify how the FMT became successfully colonized.

3. It seems that you gave FMT for only three days into the mice. It is shorter than the traditional FMT duration. Please provide a reference or evidence for that.

4. It would be clearer if you could explain why you used lecithin and maltodextrin as vehicle emulsions for your control.

5. Consistency. You mentioned eight mice per cage on page 4, line 171, but on page 7, line 321, you said you measured 6-8 animals. If you had to drop some mice, you should mention why and in which groups.

5. In figure 4 and Table 1, the plasma level of TMA and TMAO seems to indicate there are independent ways of increasing TMAO. For example, your control group had significantly lower TMA levels than the CARNITINE groups’ TMA, which made sense after treating L-carnitine. But, Figure 4.C showed that TMAO levels in both the control and the CARNITINE groups were not significant. For this argument, you explained it could be an antioxidant effect and explained AAS and GGS as an example of antioxidant functions. However, for direct pathways, you might need to measure FMO3 in the mice groups and provide this essential oil treatment that could suppress TMA oxidation by suppressing FMO3?

6. In your discussion on page 22, lines 683-689, you mentioned the NF-kB as a modulator of thrombomodulin via an anti-inflammatory function with the previous study and your parsley group; however, I could not see any of your data about NF-kB in your paper. I know you checked IFN-gamma and IL-22 in your parsley oil treatment group, which were significantly different from others. But decreasing pro-inflammatory cytokines does not mean always increasing anti-inflammatory cytokines. You need to put more references or evidence on it.

Author Response

Manuscript ID: nutrients-1945728

Status: Pending major revisions

Article type : Article

Original title: Beneficial effects of essential oils from the Mediterranean diet on gut microbiota and their metabolites in ischemic heart dis-ease and type-2 diabetes mellitus

Note that amended manuscript has been upload in the attachment.

REVIEWER 1 

Comments and Suggestions for Authors

In general, this is an essential and exciting topic that I enjoyed reading. Your team had tried a lot of experiments to provide evidence with a well-designed mice study. However, there is minor revision required with my concern.

  1. References are required on page 3, lines 112-116.

  1. Following the reviewer’s suggestion, we have included additional bibliographic references for these in the Introduction section:

[Ref.]. Bergman, E.N. Energy Contributions of Volatile Fatty Acids from the Gastrointestinal Tract in Various Species. Physiol. Rev. 1990, 70, 567–590, doi:10.1152/physrev.1990.70.2.567.

[Ref.]. Pituch, A.; et al. Butyric Acid in Functional Constipation. Prz. Gastroenterol. 2013, 8, 295–298, doi:10.5114/pg.2013.38731.

[Ref.]. Hernández, M.A.G.; et al. The Short-Chain Fatty Acid Acetate in Body Weight Control and Insulin Sensitivity. Nutrients 2019, 11, doi:10.3390/nu11081943.

  1. Additional information is needed on page 4, lines 162-167. How did you free the stool samples from patients? Did you use glycerol to freeze the stool samples?? If you are not, you need to clarify how the FMT became successfully colonized.

  1. We apologize for the lack of clarity on this point. More information has been now included in the Materials and Methods section (“2.2. Humanized Gnotobiotic Mouse Model”) as follows:

“....Then, mice received a fecal microbiota transplantation derived from a selection of patients to recolonize the gastrointestinal tract, as previously described [Ref.]. Specifically, three patients with IHD, T2DM and high plasma levels of TMAO were eligible for feces donation. Fresh fecal samples were suspended in sterile phosphate buffered saline (PBS) and glycerol was added at 10 % (v/v) final concentration; then, samples were aliquoted into cryotubes and immediately stored at -80 °C. At the moment of the transplantation, aliquot samples from these patients were thawed in ice and mixed in equal quantities as a sample pool to guarantee that every mouse received a similar microbiota load. The sample pool was centrifuged for 2 minutes at 2,800 ×g…”

  1. It seems that you gave FMT for only three days into the mice. It is shorter than the traditional FMT duration. Please provide a reference or evidence for that.

  1. There is a high variability in the literature regarding the FTM duration and we followed a FMT protocol that was previously described by other groups (e.g., Le Bastard et al., (2018), Zhang et al., (2019) and Ubeda et al., (2013)). They have shown that a FMT for three consecutive days after antibiotic treatment is enough for a successful microbiota implantation. These bibliographic references have been now included in the Materials and Methods section (“2.2. Humanized Gnotobiotic Mouse Model”).

[Ref.]. Le Bastard, Q.; et al. Fecal Microbiota Transplantation Reverses Antibiotic and Chemotherapy-Induced Gut Dysbiosis in Mice. Sci. Rep. 2018, 8, 1–11, doi:10.1038/s41598-018-24342-x.

[Ref.]. Zhang, Y.; et al. Gut Microbiota from NLRP3-Deficient Mice Ameliorates Depressive-like Behaviors by Regulating Astrocyte Dysfunction via CircHIPK2. Microbiome 2019, 7, 1–16, doi:10.1186/s40168-019-0733-3.

[Ref.]. Ubeda, C.; et al. Intestinal Microbiota Containing Barnesiella Species Cures Vancomycin-Resistant Enterococcus Faecium Colonization. Infect. Immun. 2013, 81, 965–973, doi:10.1128/IAI.01197-12.

  1. It would be clearer if you could explain why you used lecithin and maltodextrin as vehicle emulsions for your control.

  1. For oral administration, oil-in-water (O/W) emulsions were prepared with essential oils in distilled water with lecithin and maltodextrin as emulsifiers (i.e., surfactants that stabilize emulsions) to prevent the oil and water phases separation. Because essential oils were administered as O/W emulsions, a group of mice (vehicle group) was treated with the components of the emulsion but without oils to control any effect of the emulsifiers and, therefore, differences in the experimental groups.

Accordingly, the text has been revised in order to clarify the preparation of these O/W emulsions (“2.3.2. Preparation of Essential Oil Emulsions” and “2.3.3. Treatment with Essential Oil Emulsions”):

“...For oral administration, oil-in-water (O/W) emulsions were prepared with these essential oils at 3.38% (v/v) concentration in distilled water with lecithin (0.76%, w/v) and maltodextrin (21.47%, w/v) as emulsifiers to prevent the oil and water phases separation...”

“...or vehicle emulsion (i.e., treatment with a vehicle solution containing distilled water, lecithin and maltodextrin in the same proportions but without essential oils)...”

  1. Consistency. You mentioned eight mice per cage on page 4, line 171, but on page 7, line 321, you said you measured 6-8 animals. If you had to drop some mice, you should mention why and in which groups.

  1. In the Materials and Methods section, we have specified that mice were assigned to five experimental groups with eight animals per group in a cage at the beginning of the study (“2.3.1. Experimental Groups”):

“...Transplanted mice were randomly assigned to five experimental groups, with eight animals per group (n = 8) in a cage at the beginning of the study...”

As detailed in a later subsection, four mice died during the treatment period, and the number of animals per group at the end of treatments is now shown in the text (“2.3.3. Treatment with Essential Oil Emulsions”):

“...However, four mice died during the treatment period and the number of animals per group at the end of treatments was as follows: 1) control group, n = 7; 2) carnitine group, n = 7; 3) savory group, n = 8; 4) parsley group, n = 8; and, 5) rosemary group, n = 6...”

  1. In figure 4 and Table 1, the plasma level of TMA and TMAO seems to indicate there are independent ways of increasing TMAO. For example, your control group had significantly lower TMA levels than the CARNITINE groups’ TMA, which made sense after treating L-carnitine. But, Figure 4.C showed that TMAO levels in both the control and the CARNITINE groups were not significant. For this argument, you explained it could be an antioxidant effect and explained AAS and GGS as an example of antioxidant functions. However, for direct pathways, you might need to measure FMO3 in the mice groups and provide this essential oil treatment that could suppress TMA oxidation by suppressing FMO3?

  1. Effectively, TMA and TMAO levels were not found to be directly associated in our study. While there were no differences in the increased TMA levels between the groups of mice treated with L-carnitine, TMAO levels were significantly lower in the parsley and rosemary groups than in the L-carnitine group. There are different physiological/metabolic pathways to induce this decrease in the TMAO levels of mice treated with parsley and rosemary essential oils. As shown in figure (below), the metabolism of TMAO can be regulated by decreasing the oxidative activity of FMO3 and/or increasing the mobilization of TMAO (i.e., absorption by tissues or excretion in urine) from the blood.

Figure from Cho CE, Caudill MA. Trimethylamine-N-Oxide: Friend, Foe, or Simply Caught in the Cross-Fire? Trends Endocrinol Metab. 2017 Feb;28(2):121-130. doi: 10.1016/j.tem.2016.10.005.

Unfortunately, we did not determine the expression or activity of FMO3 because there were no blood samples available; furthermore, the mobilization of TMAO would have required to collect blood samples at different times after treatments with these essential oil emulsions (increasing the number of mice). We are aware that further research is needed to elucidate the mechanism of parsley and rosemary essential oils in decreasing the TMAO levels.

Accordingly, we have revised this paragraph in the Discussion section as follows:

“...Gut microbiota uses L-carnitine as a precursor to generate TMA, which is rapidly absorbed into the portal circulation by passive diffusion across the enterocyte membranes and then oxidized to TMAO by the action of hepatic flavin-containing monooxygenases (mainly FMO3) [Ref.]. As expected, our results showed that TMA was only increased after administration of L-carnitine but we did not observe changes in TMA levels after treatments with different essential oil emulsions. In contrast, we observed a significant reduction of plasma TMAO levels in the parsley and rosemary groups that cannot be explained by changes in the TMA levels. Therefore, the treatments with parsley and rosemary essential oils likely affect the metabolism of TMAO by decreasing the oxidative activity of FMO3 and/or increasing the mobilization (i.e., absorption by tissues or excretion in urine) of TMAO [Ref.]. However, the exact mechanism has to be elucidated and further research is needed....”

  1. In your discussion on page 22, lines 683-689, you mentioned the NF-kB as a modulator of thrombomodulin via an anti-inflammatory function with the previous study and your parsley group; however, I could not see any of your data about NF-kB in your paper. I know you checked IFN-gamma and IL-22 in your parsley oil treatment group, which were significantly different from others. But decreasing pro-inflammatory cytokines does not mean always increasing anti-inflammatory cytokines. You need to put more references or evidence on it.

  1. We absolutely agree with the reviewer’s comments and this paragraph has been revised and rewritten in order to clarify our discussion. We have mentioned the anti-inflammatory effects of thrombomodulin but any direct association between anti- and pro-inflammatory cytokines through the NF-kB signaling pathway has been avoided:

“...These pro- and anti-inflammatory profiles may be also related to the production of thrombomodulin, an anti-coagulant cofactor. Interestingly, recent studies have shown that thrombomodulin exhibits anti-inflammatory effects by inhibiting leukocyte recruitment [Ref.]. In addition, the prototypical pro-inflammatory NF-kB signaling pathway has been shown to down-regulate thrombomodulin expression [Ref.]; however, the elevated thrombomodulin expression in the savory and parsley groups only was associated with the decrease in the expression of the pro-inflammatory cytokine IL-22 because the expression of other pro-inflammatory cytokines was inconsistent in both groups...”

Reviewer 2 Report

Sánchez-Quintero et al’s manuscript “Beneficial effects of essential oils from the Mediterranean diet on gut microbiota and their metabolites in ischemic heart disease and type-2 diabetes mellitus” is a piece of research that holds a great amount of work, carefully planned in most aspects, strictly evaluated from a statistical point of view and meticulously carried out and, therefore, results are robust, except for a few limitations, most of them already acknowledged by the authors at the end of the Discussion section. The manuscript is carefully written, with no typos or mistakes, which allows an easy reading.

However, in spite of all this, there are a few important doubts about their work that need to be addressed, since some of them can be important when reaching conclusions.

1.     Line 152. Authors state that they use female CD1 mice. What is the purpose of using only females? Do the authors have in account the estrous cycle phase variations in their results? In our experience, if the animals are not in the same estrous cycle, it is very difficult to obtain statistically significant results with some variables.

2.     Prior to the transplantation procedure, mice were treated with ampicillin. But not all the microbiota taxa have the same sensibility to ampicillin. For instance, Tenericutes  are resistant. In fact, ampicillin is added to culture media to isolate them. And many Bacteroidetes taxa are also ampicillin resistant. Do you check somehow that microbiota has been eliminated? If residual microbiota remains, this will be different in each mouse and may affect final results in an unknown way.

3.     Line 162. [4]: I think it is in reference 31 where this procedure is described, please check.

4.     Line 163. How is the fecal transplantation performed? “A selection of patients” are chosen, but how many? Are their fecal samples mixed and this same mixture used to recolonize all mice or each individual sample is used to recolonize one individual mouse? Please, describe.

5.     Line 163. Authors indicate that samples are stored at -80ºC until used. This procedure will prevent DNA from degrading but will not prevent bacteria from lysing. It is my understanding that feces have to undergo a special treatment with cryoprotectants before being stored at -80ºC. Otherwise, the microbial composition of viable cells might have varied, even disappearing at all, and the repopulated intestinal microbiota be totally different from that of the donor’s.  This is not a real problem in this study since the control group has also the same fecal transplantation, but we have to be careful not to assume that we are reproducing the patients’ microbiota. Therefore, if the authors have followed any cryoprotectant method, please indicate so. On the other hand, if they have not, please, discuss this fact also in the Discussion section.

6.     Line 215. Authors say they use the Ion SS5TM System (Thermo Fisher Scientific) but then in figures 2 and 3, they indicate that 16S rRNA gene sequencing is performed by 454 sequencing (which used to be offered by Roche). Which one is it? 454 has been discontinued by Roche so this information might be a copy-paste error from a previous manuscript or it may be data obtained a few years ago, and in this case Material and Methods has to be changed accordingly.

7.     Lines 351-357. Please remove the residual instructions to authors.

8.     Line 409. Here it is stated that bacterial composition was compared using one-way ANOVA. However, microbiota abundance does not maintain a normal distribution and Kruskal-Wallis is used instead. In fact, authors also say so in Material and Methods, but I did not find any mention to Kruskal -Wallis later on in the Results section. Could this be a typo?

9.     Lines 627-629. “humanized mouse model harboring the colonic microbiota from patients with IHD and T2DM”:  For the reasons indicated in 5 and also to some extent 2, I do not think these mice are harboring the microbiota of the patients. This do not invalidate the results because the comparisons still prevail but only that they are not the microbiota from those patients. In fact, only by using germ-free mice this can be achieved. I think it is better to say something like “humanized mouse model harboring colonic microbiota related to/derived from that of patients with IHD and T2DM” to stay in the safe side.

10.  Line 44: “Humanized mice harboring gut microbiota from patients with IHD and T2DM” For the same reason than in 9, this sentence has to be changed in the abstract, unless authors can prove that they have reproduced the same composition. It would be “Humanized mice harboring gut microbiota related to/derived from patients with IHD and T2DM”

11.  Reference number 51 is not in the same style.

Author Response

Manuscript ID: nutrients-1945728

Status: Pending major revisions

Article type : Article

Original title: Beneficial effects of essential oils from the Mediterranean diet on gut microbiota and their metabolites in ischemic heart dis-ease and type-2 diabetes mellitus

Note that amended manuscript has been upload in the attachment.

REVIEWER 2 

Comments and Suggestions for Authors

Sánchez-Quintero et al’s manuscript “Beneficial effects of essential oils from the Mediterranean diet on gut microbiota and their metabolites in ischemic heart disease and type-2 diabetes mellitus” is a piece of research that holds a great amount of work, carefully planned in most aspects, strictly evaluated from a statistical point of view and meticulously carried out and, therefore, results are robust, except for a few limitations, most of them already acknowledged by the authors at the end of the Discussion section. The manuscript is carefully written, with no typos or mistakes, which allows an easy reading.

However, in spite of all this, there are a few important doubts about their work that need to be addressed, since some of them can be important when reaching conclusions.

  1. Line 152. Authors state that they use female CD1 mice. What is the purpose of using only females? Do the authors have in account the estrous cycle phase variations in their results? In our experience, if the animals are not in the same estrous cycle, it is very difficult to obtain statistically significant results with some variables.

  1. We understand the reviewer’s concern regarding the use of female mice. However, we used randomly cycling female mice under the assumptions that we can summarize as follows:

1) Female mammals have long been neglected or underutilized in biomedical research based on the assumption that females are intrinsically more variable than males and must be tested at each of four stages of the estrous cycle to generate reliable data.

2) In a meta-analysis of 293 articles (Prendergast, B.J.; Onishi, K.G.; Zucker, I. Female mice liberated for inclusion in neuroscience and biomedical research. Neurosci Biobehav Rev. 2014 Mar;40:1-5. doi: 10.1016/j.neubiorev.2014.01.001), behavioral, morphological, physiological, and molecular traits (30 categories) were monitored in male mice and females tested without regard to estrous cycle stage and variability was not significantly greater in females than males for any endpoint; in contrast, variability was substantially greater in males for several traits. Therefore, other studies have used randomly cycling female mice (Grant, C.V.; et al. Manipulations of the gut microbiome alter chemotherapy-induced inflammation and behavioral side effects in female mice. Brain Behav Immun. 2021 Jul;95:401-412. doi: 10.1016/j.bbi.2021.04.014).

3) The estrous cycle obviously influences behavior and physiology, but there are other sources of variability higher in male mice derived from group-housing. Specifically, group-housed males fight, establish dominance hierarchies in which the dominant male monitors and defends the entire cage space, and frequently attacks the others. These attacks active glucocorticoid and sympathetic responses in males (Emond, M.; Faubert, S.; Perkins, M. Social conflict reduction program for male mice. Contemp Top Lab Anim Sci. 2003 Sep;42(5):24-6). Unlike males, female mice housed in groups do not fight (Meakin, L.B.; et al. Male mice housed in groups engage in frequent fighting and show a lower response to additional bone loading than females or individually housed males that do not fight. Bone. 2013 May;54(1):113-7. doi: 10.1016/j.bone.2013.01.029) which was important for our study.

Nevertheless, we aware that the most informative studies are those that incorporate females and males in equal numbers with explicit comparison of the two sexes and that the inclusion of the estrous cycle stage of female mice contributes to a better characterization. Accordingly, we have justified the use of female mice in the present study in the Materials and Methods section and potential limitations derived from the experimental design have been included in the Discussion section.

  1. Prior to the transplantation procedure, mice were treated with ampicillin. But not all the microbiota taxa have the same sensibility to ampicillin. For instance, Tenericutes are resistant. In fact, ampicillin is added to culture media to isolate them. And many Bacteroidetes taxa are also ampicillin resistant. Do you check somehow that microbiota has been eliminated? If residual microbiota remains, this will be different in each mouse and may affect final results in an unknown way.

  1. We apologize for this important mistake. All-important information related to the antibiotic treatment was missed in the original version of the manuscript. In our study, mice were treated with broad-spectrum antibiotics that were administered by combining oral gavage (vancomycin, neomycin and metronidazole) and drinking water supplementation (ampicillin) during 10 consecutive days to induce gut microbiota depletion in mice based on Kennedy et al., (2018) (Kennedy, E.A.; King, K.Y.; Baldridge, M.T. Mouse Microbiota Models: Comparing Germ-Free Mice and Antibiotics Treatment as Tools for Modifying Gut Bacteria. Front Physiol. 2018 Oct 31;9:1534. doi: 10.3389/fphys.2018.01534) and Reikvam et al., (2012) (Reikvam, D.H.; et al. Depletion of murine intestinal microbiota: effects on gut mucosa and epithelial gene expression. PLoS One. 2011 Mar 21;6(3):e17996. doi: 10.1371/journal.pone.0017996). Consequently, the Materials and Methods section (“2.2. Humanized Gnotobiotic Mouse Model”) has been extensively revised to include a detailed description as follows:

“...Prior to the transplantation procedure, 18-week-old mice (n = 40) were treated with broad-spectrum antibiotics for 10 consecutive days to induce gut microbiota depletion combining both oral gavage administration and drinking water supplementation. Thus, mice were orally treated by gavage every 12 h with an antibiotic cocktail (10 mL kg-1 body weight) consisting of vancomycin (5 mg mL-1), neomycin (10 mg mL-1) and metronidazole (10 mg mL-1), and ampicillin was administered in drinking water (1 g L-1) [Ref.][Ref.]...”·

  1. Line 162. [4]: I think it is in reference 31 where this procedure is described, please check.

  1. The bibliographic reference has been accordingly changed to Le Bastard, Q. et al., 2018 (Le Bastard, Q.; et al. Fecal Microbiota Transplantation Reverses Antibiotic and Chemotherapy-Induced Gut Dysbiosis in Mice. Sci. Rep. 2018, 8, 1–11, doi:10.1038/s41598-018-24342-x).

  1. Line 163. How is the fecal transplantation performed? “A selection of patients” are chosen, but how many? Are their fecal samples mixed and this same mixture used to recolonize all mice or each individual sample is used to recolonize one individual mouse? Please, describe.

  1. For clarification a better description of the fecal transplantation procedure has been included in the Materials and Methods section (“2.2. Humanized Gnotobiotic Mouse Model”):

“...Specifically, three patients with IHD, T2DM and high plasma levels of TMAO were eligible for feces donation. Fresh fecal samples were suspended in sterile phosphate buffered saline (PBS) and glycerol was added at 10 % (v/v) final concentration; then, samples were aliquoted into cryotubes and immediately stored at -80 °C. At the moment of the transplantation, aliquot samples from these patients were thawed in ice and mixed in equal quantities as a sample pool to guarantee that every mouse received a similar microbiota load. The sample pool was centrifuged for 2 minutes at 2,800 ×g, and the supernatant was transferred to mice by oral gavage for three consecutive days according to previously published protocols [Ref.][Ref.][Ref.]. Mice were moved to clean cages every two days to minimize the coprophagia and to prevent reinoculation from old feces...”.

  1. Line 163. Authors indicate that samples are stored at -80ºC until used. This procedure will prevent DNA from degrading but will not prevent bacteria from lysing. It is my understanding that feces have to undergo a special treatment with cryoprotectants before being stored at -80ºC. Otherwise, the microbial composition of viable cells might have varied, even disappearing at all, and the repopulated intestinal microbiota be totally different from that of the donor’s. This is not a real problem in this study since the control group has also the same fecal transplantation, but we have to be careful not to assume that we are reproducing the patients’ microbiota. Therefore, if the authors have followed any cryoprotectant method, please indicate so. On the other hand, if they have not, please, discuss this fact also in the Discussion section.

  1. The reviewer is absolutely right. A poor description of the collection and preservation of fecal samples from the donors was included in the original manuscript. Please, find the correct description and clarifications in the Materials and Methods section (“2.2. Humanized Gnotobiotic Mouse Model”):

“...Specifically, three patients with IHD, T2DM and high plasma levels of TMAO were eligible for feces donation. Fresh fecal samples were suspended in sterile phosphate buffered saline (PBS) and glycerol was added at 10 % (v/v) final concentration; then, samples were aliquoted into cryotubes and immediately stored at -80 °C. At the moment of the transplantation, aliquot samples from these patients were thawed in ice and mixed in equal quantities as a sample pool to guarantee that every mouse received a similar microbiota load. The sample pool was centrifuged for 2 minutes at 2,800 ×g, and the supernatant was transferred to mice by oral gavage for three consecutive days according to previously published protocols [Ref.][Ref.][Ref.]. Mice were moved to clean cages every two days to minimize the coprophagia and to prevent reinoculation from old feces...”.

  1. Line 215. Authors say they use the Ion SS5TM System (Thermo Fisher Scientific) but then in figures 2 and 3, they indicate that 16S rRNA gene sequencing is performed by 454 sequencing (which used to be offered by Roche). Which one is it? 454 has been discontinued by Roche so this information might be a copy-paste error from a previous manuscript or it may be data obtained a few years ago, and in this case Material and Methods has to be changed accordingly.

  1. The legends for figures 2 and 3 have been revised to correct this inconsistency because sequencing was carried out on an Ion 520 chip using the Ion S5TM System (Thermo Fisher Scientific, Walthman, MA, USA).

  1. Lines 351-357. Please remove the residual instructions to authors.

  1. These residual instructions have been removed from the Materials and Methods section.

  1. Line 409. Here it is stated that bacterial composition was compared using one-way ANOVA. However, microbiota abundance does not maintain a normal distribution and Kruskal-Wallis is used instead. In fact, authors also say so in Material and Methods, but I did not find any mention to Kruskal -Wallis later on in the Results section. Could this be a typo?

  1. We apologize for this important mistake. The legend for figure 3 has been revised and the Kruskal Wallis rank-sum test has been included as the correct statistical test.

  1. Lines 627-629. “humanized mouse model harboring the colonic microbiota from patients with IHD and T2DM”: For the reasons indicated in 5 and also to some extent 2, I do not think these mice are harboring the microbiota of the patients. This do not invalidate the results because the comparisons still prevail but only that they are not the microbiota from those patients. In fact, only by using germ-free mice this can be achieved. I think it is better to say something like “humanized mouse model harboring colonic microbiota related to/derived from that of patients with IHD and T2DM” to stay in the safe side.

  1. In response to questions 5 and 2, the authors want to indicate that the fecal samples from donors were cryopreserved using PBS and glycerol and that these mice were treated with ampicillin and other antibiotics (vancomycin, neomycin and metronidazole). However, we cannot assure that these mice were germ-free because a 16S rRNA gene sequencing prior to transplantation was not performed. Therefore, we have followed the reviewer´s suggestion:

“...humanized mouse model harboring colonic microbiota derived from that of patients with IHD and T2DM...”

  1. Line 44: “Humanized mice harboring gut microbiota from patients with IHD and T2DM” For the same reason than in 9, this sentence has to be changed in the abstract, unless authors can prove that they have reproduced the same composition. It would be “Humanized mice harboring gut microbiota related to/derived from patients with IHD and T2DM”

  1. Following the reviewer’s suggestion, this sentence has been now changed in the Abstract:

“...Humanized mice harboring gut microbiota derived from that of patients with IHD and T2DM were supplemented...”

  1. Reference number 51 is not in the same style.

  1. This bibliographic reference has been revised and updated.

Round 2

Reviewer 2 Report

Thank you for addressing my comments. Congratulations on the study.

Author Response

Manuscript ID: nutrients-1945728

Status: Pending minor revisions

Article type: Article

Title: Beneficial effects of essential oils from the Mediterranean diet on gut microbiota and their metabolites in ischemic heart disease and type-2 diabetes mellitus

Journal: Nutrients

Section: Nutritional Epidemiology

Special Issue: Dietary Supplements in Cardiovascular and Metabolic Diseases

Dear Editor,

As the corresponding author (on behalf of all authors), I am writing this note in response to your comments regarding our manuscript (nutrients-1945728).

Respectfully, the authors do not agree with the inclusion of 16 additional reports after addressing satisfactorily all the points raised by the reviewers. The arguments are summarized as follows:

1) In the original version of the manuscript, there were no Editor’s comments and the Discussion was revised and rewritten according to the reviewers’ suggestions and comments. Despite the approval by the reviewers, these new comments require to change the Discussion again.

2) There is not any scientific justification or brief explanation for including all these reports in the revised manuscript. This is a long list of references (PMID numbers) that, in many cases, is difficult to integrate into the discussion.

3) The authors cannot discuss other articles/reviews but discussing the results derived from the present study. Therefore, the discussion was made on the basis of the present results using references at the discretion of the authors.

4) The authors understand the Editor can suggest some interesting articles/reviews that they didn’t notice for discussing the results; however, 16 articles are excessive (even more so when 65 articles are cited in the revised manuscript).

Therefore, we have selected four relevant references from the list and included in the manuscript without substantially modifying the revised Discussion.

Best regards

Josué Delgado